# Transcriptomic Analysis for Diurnal Temperature Differences Reveals Gene-Regulation-Network Response to Accumulation of Bioactive Ingredients of Protocorm-like Bodies in *Dendrobium officinale*

**DOI:** 10.3390/plants13060874

**Published:** 2024-03-18

**Authors:** Qingqing Chen, Chunyu Zhang, Yukun Chen, Congqiao Wang, Zhongxiong Lai

**Affiliations:** Institute of Horticultural Biotechnology, Fujian Agriculture and Forestry University, Fuzhou 350002, China; qingqing_chen777@163.com (Q.C.); zcynhba@163.com (C.Z.); cyk68@163.com (Y.C.); wcq13313983179@126.com (C.W.)

**Keywords:** *D. officinale* PLBs, diurnal temperature difference, bioactive ingredients, transcriptomic analysis, gene-regulation network

## Abstract

*Dendrobium officinale* Kimura et Migo (*D. officinale*) is one of the most important traditional Chinese medicinal herbs, celebrated for its abundant bioactive ingredients. This study demonstrated that the diurnal temperature difference (DIF) (T1: 13/13 °C, T2: 25/13 °C, and T3: 25/25 °C) was more favorable for high chlorophyll, increased polysaccharide, and total flavonoid contents compared to constant temperature treatments in *D. officinale* PLBs. The transcriptome analysis revealed 4251, 4404, and 4536 differentially expressed genes (DEGs) in three different comparisons (A: 25/13 °C vs. 13/13 °C, B: 13/13 °C vs. 25/25 °C, and C: 25/13 °C vs. 25/25 °C, respectively). The corresponding up-/down-regulated DEGs were 1562/2689, 2825/1579, and 2310/2226, respectively. GO and KEGG enrichment analyses of DEGs showed that the pathways of biosynthesis of secondary metabolites, carotenoid biosynthesis, and flavonoid biosynthesis were enriched in the top 20; further analysis of the sugar- and flavonol-metabolism pathways in *D. officinale* PLBs revealed that the DIF led to a differential gene expression in the enzymes linked to sugar metabolism, as well as to flavonol metabolism. Certain key metabolic genes related to ingredient accumulation were identified, including those involved in polysaccharide metabolism (*SUS*, *SUT*, *HKL1*, *HGL*, *AMY1*, and *SS3*) and flavonol (*UGT73C* and *UGT73D*) metabolism. Therefore, these findings indicated that these genes may play an important role in the regulatory network of the DIF in the functional metabolites of *D. officinale* PLBs. In a MapMan annotation of abiotic stress pathways, the DEGs with significant changes in their expression levels were mainly concentrated in the heat-stress pathways, including heat-shock proteins (HSPs) and heat-shock transcription factors (HSFs). In particular, the expression levels of *HSP18.2*, *HSP70*, and *HSF1* were significantly increased under DIF treatment, which suggested that *HSF1*, *HSP70* and *HSP18.2* may respond to the DIF. In addition, they can be used as candidate genes to study the effect of the DIF on the PLBs of *D. officinale*. The results of our qPCR analysis are consistent with those of the transcriptome-expression analysis, indicating the reliability of the sequencing. The results of this study revealed the transcriptome mechanism of the DIF on the accumulation of the functional metabolic components of *D. officinale*. Furthermore, they also provide an important theoretical basis for improving the quality of *D. officinale* via the DIF in production.

## 1. Introduction

*Dendrobium officinale* Kimura et Migo (*D. officinale*), also known as one of the ‘the first of nine Chinese fairy grasses’ [1], is an economically important herb used in traditional Chinese medicine [2]. *D. officinale* contains various active components, including polysaccharides, amino acids, alkaloids, flavonoids, terpenes, phenols, coumarins, terpenoids, phenyl compounds, and other several essential microelements [1,2,3,4,5,6]. The ingredients extracted from *D. officinale* have been reported to be beneficial for one’s health. For example, dendrobium polysaccharides, the main bioactive ingredient composed of mannose and glucose can ameliorate hyperglycemia, alleviate type 2 diabetes, regulate the immune response, and exhibit antitumor and anti-inflammatory activities [3,7,8]. Dendrobium flavonoids are known for their antioxidant and protective cytotoxicity [9,10], and dendrobium alkaloids have anti-inflammatory, antipyretic, eye-benefitting, and immune-regulatory effects [2,11].

Wild growth or field planting is not economically feasible due to requiring a high quantity of time, resources, and labor costs. Using a protocorm-like body (PLB) is a productive method for harvesting a large number of PLBs in a short cultivating period when using a culture medium [12]. Therefore, the accumulation and source of bioactive ingredients are critical for the resource-sustainable use of *D. officinale*.

Currently, most studies have shown that the diurnal temperature difference (DIF) can regulate plant growth and development, aiding in improving internode length, plant height, stem thickness, the number of leaves, leaf area, and dry-matter accumulation [2,12,13].

It has been suggested that a lower DIF favors the morphological growth of test-tube seedlings, while a higher DIF is required for soluble sugar accumulation [14]. The study by Yang Bo et al. on Gerbera test-tube seedlings also showed that a large DIF (9 °C) favors the accumulation of soluble sugars and dry matter in test-tube seedlings [15]. The expression level of the *DoGAUT1* gene increased linearly in *D. officinale* under the day/night 25/15 °C temperature-variation treatment when compared to a constant 25 °C treatment, synthesizing large amounts of extracellular polysaccharides, which may act to protect *D. officinale* from low-temperature damage [16].

Changes in temperature stimulate the response of certain genes, causing changes in the metabolic pathways involved, which in turn affect plant growth and development and the accumulation of functional metabolites. However, the impact of the DIF on polysaccharide and secondary metabolism for *D. officinale* PLBs is still absent.

Therefore, this study aimed to investigate the effect of DIF treatment on the functional ingredients in *D. officinale* PLBs using the transcriptome and real-time quantitative polymerase chain reaction (qRT-PCR) technique. After a constant-temperature (25/25 °C; day/night) treatment for 30 days, three temperature combinations (T1: 13/13 °C, T2: 25/13 °C, and T3: 25/25 °C) were used for 5 days to reveal the differentially expressed genes (DEGs) and their corresponding physiological pathways, in which the mining of the key candidate genes involved in the temperature-signaling network was preformed.

## 2. Results

### 2.1. DIF’s Affects on Functional Metabolites in D. officinale PLBs

The DIF affected the content of polysaccharides, alkaloids, and total flavonoids in PLBs of *D. officinale*. As shown in Figure 1A, the highest polysaccharide content of 37.71% was found in T2, followed by 23.49% in T3, and the lowest polysaccharide content of 22.56% was found in T1 (Figure 1A). Therefore, the DIF treatment was more favorable for the increase in the soluble polysaccharide content in *D. officinale* PLBs compared with the two constant-temperature treatments.

Figure 1B shows that the temperature had no significant effect on the alkaloid content in the PLBs of *D. officinale* when compared with a constant-temperature treatment, but T2 was lower than T1 and T3, indicating that a temperature-variation treatment is not conducive to an increase in alkaloid content when compared with the two constant-temperature treatments. As shown in Figure 1C, the flavonoid content (232.24 μg/g) in the T2 treatment was higher than that in T1 (172.6 μg/g) and T3 (141.38 μg/g), indicating that the DIF was favorable for the accumulation of flavonoids.

### 2.2. Analysis of the Effect of the DIF on the Physiological Indices of D. officinale PLBs

In our previous study, it was found that *D. officinale* PLBs grow slowly, and a 5-day treatment had no significant effect on the growth amount. We measured the photosynthetic pigments and antioxidant enzyme activities to study their physiological changes.

The temperature-treatment results showed that the chlorophyll a (Chla), chlorophyll b (Chlb), and chlorophyll a + b (Chla + b) levels were significantly higher in the T2 treatment than in the two constant-temperature treatments (Figure 1E). The T2 treatment saw the lowest ratio of chlorophyll a and chlorophyll b (Chla/b) (Figure 1D), while the carotenoid (Cx) content was the highest in the T1 treatment (which followed the order of T1 > T2 > T3). However, the difference in Cx content between the T1 and T2 treatments was not significant (Figure 1E).

The activity of the superoxide dismutase (SOD) and peroxidase (POD) in the T1 treatment with an all-day low temperature was significantly higher than in the other two temperature treatments. The SOD activity in *D. officinale* PLBs under the T1 treatment was 2.35 and 1.72 times higher than under the T2 and T3 treatments, respectively. POD enzyme activity under T1 treatment was 2.79 times higher than under the T2 treatment and 3.2 times higher than under the T3 treatment. SOD and POD activities under the T2 treatment were not significantly different from those under the T3 treatment. The malondialdehyde (MDA) values under the T2 treatment had a DIF that was higher than the other two constant-temperature T1 and T3 treatments. None of the differences in the MDA values were significant when comparing the T3 treatment with the T1 and T2 treatments, but there was a significant difference in the MDA values between treatments T1 and T2.

### 2.3. Data Filtering and Reference-Sequence Alignment in D. officinale PLBs

#### 2.3.1. Analysis of Transcriptome-Sequencing Basic Data

This study constructed a strand-specific cDNA library of *D. officinale* PLBs obtained from different temperatures (i.e., T1:13/13 °C, T2:25/13 °C, T3:25/25 °C) to analyze their metabolism. The libraries of mRNAs from the three different temperature-treated PLBs of *D. officinale* were sequenced using the Illumina HiSeq platform, and an average of 6.67 Gb of data was obtained for each sample. The statistics of sequencing-data quality are presented in Table 1. The gene numbers of the three cases of FPKM were counted using StringTie software (v1.0.4, http://ccb.jhu.edu/software/stringtie, accessed on date 11 March 2024) to compare the changes in gene expression in the PLBs of *D. officinale* under different temperature treatments. The statistical results showed that the number of genes with medium and high expression levels was the highest, with 12,894, 12,433, and 12,236 genes in T1, T2, and T3, respectively (Figure 2).

#### 2.3.2. Numbers of DEGs

To further investigate the transcriptome changes in the PLBs of *D. officinale* under different temperature treatments, a differential detection was performed using the PossionDis method [17]. The gene expression of T2 was compared with T1 and set as Comparison A. Similarly, T1 and T3 were set as Comparison B, and T2 and T3 were set as Comparison C. A total of 4251 (Comparison A), 4404 (Comparison B), and 4536 (Comparison C) differently expressed genes (DEGs, fold change ≥ 2.00, and FDR ≤ 0.01) were screened. There were 1562 (Comparison A) and 2310 (Comparison C) DEGs with an up-regulated expression in the respective comparisons, and the numbers of down-regulated expressions were 2689 (Comparison A), and 2226 (Comparison C), respectively; there were also 28,126 (Comparison A) and 28,021 (Comparison C) no-significant difference genes (No-DEGs), respectively (Figure 3).

A Venn diagram analysis was performed on the differential genes in Comparison A, Comparison B, and Comparison C. Among them, 1267 (459 up-regulated/808 down-regulated) DEGs only changed their expression levels in Comparison A, indicating that they were specifically responsive to daytime DIF, while 1304 (620 up-regulated/684 down-regulated) DEGs only changed their expression levels in Comparison C, indicating that they were specifically responsive to night-time DIF. A total of 1348 (496 up-regulated/852 down-regulated) DEGs show changes in their expression levels both in Comparisons A and C, indicating that they respond specifically to the DIF throughout the day. In Comparison A and Comparison C, the DEGs repeated in Comparison B were excluded, and there were 3919 DEGs with specific responses to the DIF (Figure 4).

#### 2.3.3. Gene Ontology (GO) Analysis of DEGs

A GO enrichment analysis was, respectively, performed for Comparisons A, B, and C to further analyze the changes in the transcriptome in the PLBs treated with different temperatures (Figure 5, Appendix A). The differential genes were annotated into three major functional classes: molecular function (MF), cellular component (CC), and biological process (BP). In terms of biological processes, the three main functional types with the highest number of DEGs in the DIF Comparisons A and C were the same: the metabolic process (676 and 583 genes), cellular process (620 and 561 genes), and single-organism process (498 and 461 genes). In the classification of cellular components, the top three numbers of DEGs in Comparison A and Comparison C were in the following aspects: cell (560 and 455 genes), cell part (558 and 451 genes), and membrane (423 and 415 genes). In the classification of molecular function, the top three functional types that enriched the greatest number of DEGs in the DIF Comparisons A and C were catalytic activity (713 and 639 genes), binding (614 and 516 genes), and transporter activity (78 and 79 genes) (Figure 5A,C). In conclusion, the GO functional enrichment of DEGs was basically the same in DIF Comparisons A and C. However, the number of DEGs in Comparison A was basically greater than that in Comparison C, and more DEGs were down-regulated than up-regulated (Figure 5B,D).

#### 2.3.4. Kyoto Encyclopedia of Genes and Genomes Pathway (KEGG) Analysis

Based on the results of differential gene detection, this study performed a KEGG biological-pathway classification, as well as an enrichment analysis of the DEGs. Based on the KEGG database, a total of 29,707 genes were annotated into the KEGG. In two comparisons of the DIF, i.e., Comparisons A and C, there were 3112 and 3322 DEGs that were matched to 131and 130 KEGG metabolic pathways, respectively.

The genes could be classified into seven branches according to the KEGG metabolic pathway in which they were involved, and the DEGs for which KEGG annotations were obtained were involved in five of these branches (cellular processes, environmental information processing, genetic information processing, metabolism, and organismal systems). In the metabolism branch, there were 2379 and 2510 genes annotated to 11 types of metabolic pathways in DIF Comparisons A and C, respectively. Among them, those involving the highest number of differential genes were all in the global and overview maps. This was followed by the carbohydrate metabolism and, again, the metabolic pathway type, which differed slightly between comparisons (i.e., in Comparison A, it was the amino acid metabolism, while in Comparison C, it was the biosynthesis of other secondary metabolites) (Figure 6A,C).

In order to gain a better understanding of the effect of temperature on the metabolic pathways in *D. officinale* PLBs, the top 20 most significantly enriched KEGG pathways were obtained, according to the q-value and rich factor (Figure 6B,D and Appendix A). Three pathways, as well as the secondary metabolite synthesis, carotenoid biosynthesis, and flavonoid biosynthesis, were enriched to reach the top 20 positions in the three comparisons (Appendix A). This suggested that these three pathways are responsive to temperature. The three pathways of ABC transporters, Diterpenoid biosynthesis, and ribosome generation in eukaryotes were enriched to the top 20 in both Comparisons A and C. These metabolic pathways are involved in transmembrane material transport, protein synthesis, and alkaloid synthesis in plants. The results indicated that the DIF played an important role in the synthesis of proteins and alkaloids, as well as the transmembrane transport of substances, compared with the whole-day constant temperature. Certain pathways were only enriched in the top 20 in one Comparison, such as photosynthesis, carbon fixation in photosynthetic organisms, glycosphingolipid biosynthesis–globoseries, etc., and they were only enriched in the top 20 in Comparison A, thus suggesting that the daytime temperature changes affects photosynthesis, as well as the resistance of *D. officinale* PLBs. Certain pathways were enriched in the top 20 in the Comparison C, such as the pentose phosphate pathway, fructose and mannose metabolism, isoflavone biosynthesis, etc. It has been suggested that changes in night-time temperature play a specific role in influencing sugar metabolism and the synthesis of secondary metabolites such as flavonoids in *D. officinale* PLBs (Appendix A).

#### 2.3.5. KEGG Analysis of DEGs in DIF-Specific Response

The DEGs co-expressed in Comparison B were filtered in Comparison A. The remaining DEGs responded specifically to the DIF, especially to the DIF during the daytime. These DEGs were subjected to a KEGG enrichment analysis, and DNA replication was the most enriched pathway, followed by mismatch repair, Diterpenoid biosynthesis, plant–pathogen interaction, protein export, fructose and mannose metabolism (Figure 7A).

The DEGs co-expressed in Comparison B were filtered in Comparison C. The remaining DEGs also responded specifically to the DIF, mainly in response to the night-time DIF. These DEGs are mainly enriched in Glutathione metabolism, fructose and mannose metabolism, Basal transcription factors, Diterpenoid biosynthesis, Stilbenoid, diarylheptanoid and gingerol biosynthesis, and the MAPK-signaling pathway in the plant (Figure 7B).

After integrating the DEGs in Comparison A and Comparison C, and filtering the co-expressed DEGs in Comparison B, the remaining DEGs contained all the DIF-specific response DEGs. The pathways that were significantly enriched for these DEGs were the plant–pathogen interaction, followed by DNA replication, mismatch repair, flavonoid biosynthesis, and fructose and mannose metabolism. These were followed by Basal transcription factors and protein export (Figure 7C).

The DIF mainly induced changes in pathways related to plant-damage repair and resistance enhancement, such as the plant–pathogen interaction, DNA replication, mismatch repair, and glutathione metabolism. Some pathways related to functional metabolites such as flavonoid biosynthesis, fructose and mannose metabolism, diterpenoid biosynthesis, etc., had also undergone significant changes.

### 2.4. DEGs Annotated by MapMan in D. officinale PLBs

To comprehensively understand the pathway variation of *D. officinale* PLBs under temperature treatments, a MapMan annotation was applied to analyze the DEGs. The results of the DEGs are shown in Figure 8. In the DIF Comparisons, the DEGs annotated for Comparisons A and C were 3495 and 3841, respectively. These genes were found to be mainly related to the secondary metabolic pathways, including the cell wall, lipids, light reactions, tetrapyrrole, oxidative pentose phosphate (OPP), tricarboxylic acid (TCA), ascorbate, glutathione, flavonoids, terpenes, phenylpropanoids, and phenolics (Figure 8).

The analysis of the DEGs in Comparison A exhibited that the DNA, hormone metabolism, RNA, and biotic stress-related genes were enriched (Appendix A). The top two enriched pathways were all related to DNA synthesis/RNA structure. Histone (DNA synthesis/chromatin structure histone) can affect plant growth and development, maturation and senescence, and stress response through ubiquitination, methylation, phosphorylation, and acetylation. Gene enrichment occurred in the hormone response annotated to the pathways of ABA metabolism, GA synthesis, late GA synthesis, and jasmonic acid synthesis, which are closely related to plant growth and resistance (Appendix A).

In Comparison C, the DEGs were significantly enriched in the pathways related to secondary metabolism, the RNA regulation of transcription, the WRKY domain transcription factor family, hormone response, abiotic stress, lipid metabolism, etc. (Appendix A). Therefore, we hypothesized that the DIF caused by night-time temperature changes may cause changes in transcription factors such as WRKY, which in turn regulate the accumulation of secondary metabolites and adaptation to the environment.

### 2.5. The DEGs Related to Polysaccharide Metabolism

The comparisons between different daytime/night-time temperature treatments are shown in Figure 9A, and Appendix A displays the sucrose anabolism-related DEGs subjected to different temperature treatments. Among these, a 3.076-fold increase in the sucrose synthase gene 4 (*SUS4*) expression level and a 2.47-fold decrease in the alkaline/neutral invertase D (A/N-INVD, Inv) expression were observed in Comparison A. In Comparison C, one sucrose transporter gene (*SUT2*) and one Inv gene expression level were up-regulated, while another 3 SUT genes (two *SUT2s* and one *SUT4*) and one hexokinase-like gene (*HKL1)* were found to be down-regulated. Among the differentially expressed genes with a significant-fold change in expression, *SUS4* (dendrobium_glean_10105017) was significantly up-regulated only in Comparison A, and *SUT2* (pequ_07166-d3) and *HKL1* (dendrobium_glean_10013068) were significantly down-regulated only in Comparison C. Presumably, they may be candidate genes for the DIF response.

The starch metabolism-related DEGs in *D. officinale* PLBs varied in the different temperature-treated Comparisons (Figure 9B, Appendix A). In these DEGs, which specifically responded to DIF, heteroglycan glucosidase (*HGL1*, dendrobium_glean_10117747) and alpha-amylase-like (*AMY1*, dendrobium_glean_10026401) changed expression only in Comparison A, suggesting that they may respond to the DIF during the day. Starch synthase 3 (*SS3*, dendrobium_glean_10016137) was found to have a significantly up-regulated expression only in Comparison C, indicating that the DIF caused by the low temperature at night may affect amylose formation. *HGL1* (dendrobium_glean_10117749) expression was significantly down-regulated by −1.65-fold and −2.19-fold only in Comparison A and Comparison C, respectively, suggesting that they respond to the DIF throughout the day (Appendix A).

### 2.6. The DEGs Related to Secondary Metabolism

The analysis of the DEGs in the secondary metabolism revealed that the differences were between the two DIF comparisons, A and C, in the pathways of phenlypropanoid, lignin and lignans, flavonoids, terpenoids, carotenoids, and alkaloid-like components (Figure 10). The main secondary medicinal metabolites of *D. officinale* are flavonoids and alkaloids. In our study on the effect of the DIF on the production of functional metabolites in *D. officinale* PLBs, we found that the DIF did not significantly affect the alkaloid content of the PLBs, but it did significantly affect the accumulation of flavonoids. The flavonoid pathways including chalcones, isoflavonoids, dihydroflavonds, anthocyanins, and flavonols were analyzed, and the flavonol metabolism was found to be the most diverse in differentially expressed genes (Figure 10).

UDP-glucosyl transferase (*UGTs*) obtained the largest variation in the flavonol pathway (Appendix A). For example, eight of the UGTs were up-regulated and three of them were down-regulated in Comparison A; ten of the UGTs were down-regulated, and five of them were up-regulated in Comparison B; and ten of the *UGTs* were down-regulated, and eight of them were up-regulated in Comparison C. The 22 *UGTs* whose expression was modified by changes in temperature were mainly members of the *UGT73* family. The expression of gain-of-function in ABA-modulated seed germination gene (*GAS2*, dendrobium_glean_10060927), *UGT73B4* (dendrobium_glean_10132494), and *UGT73C1* (dendrobium_glean_10067948) were significantly altered only in Comparison A, suggesting that they may have a specific response to the DIF in the daytime. *UGT73C5* (dendrobium_glean_10022776), *GAS2* (dendrobium_glean_10021516, dendrobium_glean_10124444), *UGT73B3* (dendrobium_glean_10052234, dendrobium_glean_10132500), *UGT73C6* (dendrobium_glean_10079736), and flavanone 3 hydroxylase (*F3H*, dendrobium_glean_10080291) changed expression only in Comparison C, suggesting that they may have a specific response to the nocturnal DIF. The expression of some DEGs changed in both Comparisons A and C. These included the expressions of *UGT73B5* (dendrobium_glean_10132499), *UGT73C6* (dendrobium_glean_10081750), *UGT73D1* (dendrobium_glean_10084962), and *UGT73D1* (dendrobium_glean_1008 4968), as they changed significantly in only two Comparisons, A and C, and three of these showed a significant increase in expression. Comparisons A and C are the Comparisons between the DIF treatment and the two constant-temperature treatments, respectively. The results of this Comparison suggested that these four genes may respond specifically to the DIF, whether day or night.

### 2.7. The Impact of the DIF on DEGs Related to Heat Stress

Significant changes were also observed in the expression of genes involved in abiotic stress, photosynthesis, and the biological-clock response under the influence of temperature variation. The abiotic-stress DEGs were mainly enriched in the heat-stress pathway. The PLBs of *D. officinale* mainly responded to temperature changes through alterations in the DEGs on the heat-stress pathway, which included the heat-shock protein genes (*HSPs*) and the heat-shock transcription factor genes (*HSFs*) family. These also included the heat-shock protein *HSP98.7* (dendrobium_glean_10031800), *HSP18.2* (dendrobium_glean_10087095), and *HSP70* (dendrobium_glean_10045076), as well as the heat-shock transcription factor *HSF1* (dendrobium_glean_10092271, *HSFA1A*). It was found that *HSF3* (dendrobium_glean_10092269) and *HSFA2* (dendrobium_glean_10039865) were only significantly differentially expressed in Comparisons A and C, thus suggesting that they are involved in the response to the DIF. Among them, the DEGs with the most significant differential expression folds were *HSP18.2* (dendrobium_glean_10087095) and *HSF1* (dendrobium_glean_ 10092271). Dendrobium_glean_10087095 showed a 9.29- and 4.38-fold increase in expression in Comparisons A and C, respectively. Dendrobium_glean_10092271 showed a 2.99- and 8.55-fold increase in expression in Comparisons A and C, respectively. These results suggested that they may be especially sensitive to the DIF (Figure 11, Appendix A).

### 2.8. Verification and Expression Analysis of the Key DEGs in D. officinale PLBs

According to the transcriptomic data of *D. officinale* PLBs, it was confirmed that there were multiple response pathways under the temperature variation treatment, which played an important role in the accumulation of the functional metabolites of *D. officinale* PLBs. The key genes of the *D. officinale* PLBs’ response to temperature variation were mined, and 18 DEGs related to the signals network form temperature variation were selected for qPCR verification (Table 2). Among these 18 DEGs, 16 genes were confirmed, and the other 2 genes were not capable of being amplified. These DEGs included the pyridoxal phosphate (PLP)-dependent transferases superfamily protein (*POP*); cytochrome P450 (*CYP71B24*); cysteine-rich receptor-like protein kinase (*CRK8*); coumarin synthase (*COSY*); cinnamyl alcohol dehydrogenase 6 (*CAD6*); N-acetylserotonin O-methyltransferase (*ASMT*); zeitlupe (*ZTL*); *SUS4*; *SS3*; photosystem I P700 chlorophyll a apoprotein A2 (*PSAB*); *HSP18.2*; *HSF1*; photosystem II reaction-center protein a (*PSBA*); hydroxyphenylpyruvate reductase2 (*HPPR2*); glycine cleavage system H protein 3 (*GLDC*); and a ferredoxin-like protein. These were associated with the pathways such as starch synthesis, pentose phosphate, and heat-stress resistance, as well as phenylpropanoid, lignin and lignans, flavonoids, and terpenoids, etc. Consistent with the transcriptomic results, the expression levels of *POP*, *CAD6, ASMT*, *SUS4*, *SS3*, *HSP18.2*, *HSF1*, *PSBA*, *PSAB*, and *GLDC* at different diurnal temperature samples were higher than that of the two constant-temperature treated samples, while the *CYP71B24*, *CRK8*, *ZTL*, *HPPR2*, and the ferredoxin-like protein were found to be lower (Figure 12). The qPCR results showed that the trend of gene expression changes was approximately the same as the transcriptome-sequencing results under the DIF treatments.

## 3. Discussion

### 3.1. The DIF Affects the Physiological State in D. officinale PLBs

Chlorophyll, the main photosynthetic pigment in plants, with its central part, the porphyrin ring, play a crucial role in trapping light energy [18]. Consequently, the plant’s chlorophyll content directly influences the utilization of light energy during photosynthesis [19]. Cockshull [20] found that the chlorophyll content of chrysanthemum leaves decreased significantly as the DIF fell. Similarly, Chl a and Chl b in tomatoes were found to be significantly increased under a positive DIF; however, the Chl a/b ratio decreased [21]. In agreement with this, our study revealed that chlorophyll a, chlorophyll b, and total chlorophyll content were highest under the DIF. Lower Chl a/b values enhanced the plant’s use of light energy in low-light environments. Ma [22] studied the light adaptation of Camptotheca acuminata, whereby after a reduction in the Chla/b ratio, it was observed under shade treatment. This was found to be mainly due to the increase in Chl b content, which may contribute to the enhancement in the light capture that is performed for photosynthesis. In our study, despite increased Chl a and Chl b contents under the DIF treatment, the greater increase in Chl b content resulted in the lowest Chl a/b ratio. Therefore, we suggest that *D. officinale* PLBs can enhance light-energy utilization and promote photosynthesis in variable temperature environments by increasing chlorophyll content, particularly Chl b.

Under stress, plants produce large quantities of reactive oxygen species, disrupting the balance of reactive oxygen species’ metabolism; this, in turn, can damage the membrane system and intracellular tissues, thus affecting normal plant growth and development [23]. Superoxide dismutase SOD catalyzes superoxide anions, contributing to the production of H2O2 and O2. As a result, SOD traps superoxide anions and plays a crucial role in biological antioxidant systems by generating a powerful oxidant, H_2_O_2_ [24]. Peroxidase (POD) catalyzes the oxidation of phenolic and amino compounds, eliminating hydrogen peroxide and the toxicity of phenolic and amino compounds [25]. In our study, SOD and POD activities in *D. officinale* PLBs were significantly higher under the T1 treatment (13/13 °C) than under the T2 (25/13 °C) and T3 (25/25 °C) treatments. These results suggest that low temperatures can increase SOD and POD activity, mitigating damage caused by substances such as hydrogen peroxide produced at low temperatures, thereby maintaining normal growth and development. Malondialdehyde (MDA), the end product of lipid membrane peroxidation, can deactivate proteins and nucleic acids, causing cell and membrane damage. MDA accumulation in plants reflects the degree of damage [26]. Under the DIF and 25/25 °C treatments, MDA values were higher than under the low-temperature treatment (13 °C). This may be due to the fact that PLBs reduce cell damage at low temperatures by increasing the enzymes activities, such as SOD and POD, improving the resistance of PLBs.

### 3.2. The DIF Promotes the Accumulation of Major Functional Components in in D. officinale PLBs

Polysaccharides, the main functional component of *D. officinale*, are crucial for assessing its quality [27]. Temperature affects various metabolic activities in plant cells. Within a certain temperature range, intracellular hydrolase activity increases with rising temperatures, accelerating starch hydrolysis and increasing the soluble sugar content. Conversely, as temperature falls, respiration weakens and metabolic activity decreases, reducing sugar consumption, and promoting sugar accumulation. In our study, the DIF favored the accumulation of soluble polysaccharides in the *D. officinale* PLBs. These results are similar to those of Yang Zaiqiang et al. [28]. Yuan et al. [29] explored the effect of the DIF on tomato-fruit morphology and quality attributes, finding increased soluble sugars under a positive DIF.

Flavonoids, used as indicators for *D. officinale* quality [30], play vital roles in various plant processes, such as morphogenesis, energy transport, photosynthesis and respiration, and the regulation of growth hormones and growth regulators, as well as stress resistance. Ya Liu et al. [31] found that the contents of primary and secondary metabolites, including soluble sugar, starch, as well as total phenols and flavonoids in *A. membranaceus* and *C. lanceolata* plug seedlings, were significantly enhanced with an increased DIF. This is consistent with our results. The flavonoid content of *D. officinale* PLBs in the DIF treatment was significantly higher than in the other two thermostatic treatments.

Tall fescue produced more alkaloids to adapt to elevated temperatures and drought stress [32]. Temperature affects different types of alkaloids differently. High temperatures could promote the accumulation of different alkaloids in *C. roseus*. However, in a long-term experiment, concentrations of monomeric alkaloids, such as catharanthine and vindoline, were found to be higher at 20 °C than at 25 °C; as well as this, we saw a sharp increase under 35 °C [33]. In our study, a 5-day, all-day low temperature (13/13 °C) slightly increased alkaloids’ accumulation in *D. officinale* PLBs compared to the DIF and control treatments (but the effect was not significant).

Due to the high polysaccharide content in *D. officinale*, this was found to be the most dominant functional product. Therefore, we concluded that the DIF favored the accumulation of main functional products in the *D. officinale* PLBs.

### 3.3. The Polysaccharide Metabolism Process in Response to Temperature Change in D. officinale PLBs

Sucrose synthase (Sus) is a key enzyme in sucrose metabolism and plant growth, responsible for converting UDP (uridine diphosphate) and sucrose into UDP-glucose and fructose [34]. The *SUS* gene encodes the sucrose synthase enzyme and is present in a variety of plants [35,36]. *SUS* also positively influences stress tolerance, with low temperature and drought stress inducing the expression of the *SUS* genes in *Hevea brasiliensis* (para rubber tree) [37]. It has been shown that, in *S. tuberosum*, the *SUS4* gene is mainly expressed in developing tubers, and it also affects the starchiness in tubers [38]. The down-regulation of cucumber sucrose synthase 4 (*CsSuSy4*) resulted in low supplies of hexose, starch, and cellulose, which inhibited flower and fruit growth and development [39]. The overexpression of *StSUS4* enhanced the accumulation of starch and ADP-glucose in the maize seed endosperms by 10–15% [39]. Our study indentified two *SUS4* genes responding to the DIF (25/13 °C). Particularly, dendrobium_glean_10105017 was significantly up-regulated by a factor of 3.08 in Comparison A. Previous studies found *SUS4* expression in Saccharum species responding to diurnal cycles [40]. This suggested that the significant up-regulation of dendrobium_glean_10105017 might be related to an increased daytime temperature, which would subsequently affect the polysaccharide metabolism in *D. officinale* PLBs.

The sucrose-transporter (*SUT*) gene is the main carrier of active sucrose transport [41]. *SUT* family members exhibit differential expressions based on organ or tissue specificity, and various biotic or abiotic stresses [42]. In our study, only *SUT2s* and *SUT4s* responded to the DIF. *OsSUT2* plays a crucial role in regulating photosynthesis products, and it also increases the seed setting rate and grain filling rate of rice through the transport of photosynthetic products [41]. *PsSUT2* in Paeonia lactiflora not only accelerated nutrient growth and sucrose accumulation in the sink organ but may also be involved in the signal transduction process during flowering, thereby bringing forward the time of flowering [43]. The knockdown of *OsSUT4* inhibited the accumulation of sucrose and starch in the leaves of mutant rice, due to the blockage of photosynthate loading at the source end, as well as due to the insufficient supply of sucrose at the sink end, which resulted in inhibition in the photosynthesis rate, delayed grain formation, and a decrease in the quality of the grains [41]. The expression level of *OsSUT4* in germinating embryos and mature pollen was regulated by temperature: the higher the temperature, the higher the expression of the *OsSUT4* in the embryos and pollen [44]. The up-regulation of *SUC2* and *SUC4* expression was found in the *SUT* family genes in *Arabidopsis* when subjected to low-temperature stress [45]. However, the study by Bilska-Kos et al. [46] found that the expression of the *SUT2* gene in maize underwent a down-regulated expression after 28 h of low-temperature treatment. It is thought that *SUT2* may not play a major transport role in this process, but that it may rather function as an element that regulates *SUT4* [46,47]. It is also possible that because the expression of sucrose-transporter proteins correlates with the current concentration of sucrose in the cell, high levels of sugar in the cell may lead to the inhibition of this transporter [46]. Our study found different *SUT2s* responded differently to the DIF. Notably, the expression of the *SUT2* gene (pequ_07166-d3) was significantly down-regulated by 3.12 fold in Comparison C. We speculated that this gene may primarily respond to the DIF caused by decreased night-time temperatures, and *SUT2s* and *SUT4s* may regulate the sucrose-transport pathway in *D. officinale* PLBs by interacting under a DIF.

Starch synthase (SS) is the key enzyme in starch synthesis, catalyzing the transfer of the glucosyl moiety from ADP-glucose to the non-reducing end of elongated glucan chains for the purpose of amylopectin synthesis [48]. Our study found two SS genes with significantly changed expression under the influence of temperature: *SS3* (dendrobium_glean_10016137) and *SS4* (dendrobium_glean_10116809). *SS3* plays an important role in branched starch synthesis and is responsible for the synthesis of branched starch B1 and B2 chains [49]. *SS4* is associated with starch-granule-initiation formation [50]. The expression of *CsSS1*, *CsSS3,* and *CsSS4* tended to be up-regulated during the cold domestication of tea trees [51]. The expressions of *SSSIIa* and *SSSIIIa* in developing rice endosperms were reduced at high temperatures [52]. This is similar to our findings. In our study, the expression of *SS3* was significantly up-regulated in Comparison C, possibly due to the DIF caused by a decreased night-time temperature.

### 3.4. Secondary Metabolic Processes in Response to DIF in D. officinale PLBs

*UGTs* are involved in various stress responses [53]. Our study identified a large number of uridine diphosphate glucuronosyltransferases (*UGTs*) in the flavonol pathway, whose expression changed with temperature variations. The *UGT73* family, closely related to the secondary metabolism in plants, was prominently affected. *UGT73B5*, *UGT73C6*, and 2 *UGT73D1* respond to the DIF throughout the day. *AtUGT73B5* responded to H_2_O_2_ and *Pseudomonas syringae* [54]. Jones et al. [55] found that *AtUGT73C6* was involved in flavonol glycoside’s biosynthesis in *Arabidopsis thaliana*; it catalyzes the glucose from UDP-glucose to the 7-OH position of kaempferol-3-O-rhamnoside and quercetin-3-O-rhamnoside. Most members of UGT73D group in *Arabidopsis thaliana* were related to the synthesis of flavonoids, rapesolactones, and plant defense [56]. The *AtUGT73D1* gene can be induced by SA but not MeJA [57]. The expression of *CsUGT73D1* can be induced by various plant hormones such as gibberellic acid (GA), abscisic acid (ABA), and methyl jasmonate (MeJA), playing a pivotal role in the regulation of multiple hormone-signaling pathways and exerting diverse effects on plant growth and development [58]. The other *UGT73s* have also been shown to display flavonol glycotransferase activities. In our study, we found that the DIF had the greatest effect on *UGTs* in the flavonol pathway of *D. officinale* PLBs. There were different change expressions for *UGT73* family genes, thus suggesting that different *UGTs* respond differently to a DIF. The DIF may affect different hormone-signaling pathways by regulating changes in different *UGT73* family genes, such as *UGT73B5*, *UGT73C6,* and *UGT73D1*, which in turn may have an effect on the resistance and growth of *D. officinale* PLBs.

### 3.5. Genes Involved in the Heat-Stress Pathway Play an Important Role in D. officinale PLBs’ Response to DIF

*D. officinale* PLBs respond to low temperatures mainly through changes in DEGs in the heat-stress pathway. The primary genes involved are the heat-stress protein (*HSPs*) and heat-stress transcription factor (*HSFs*) families. HSPs are a class of proteins that are activated and produced in large quantities in plants and animals in response to abiotic stresses such as humidity, temperature, salinity, heavy metals, and ultraviolet light [59]. *HSFs* are heat-stimulated transcription factors that can bind to the *HSE* promoter element in *HSPs* and regulate the opening and closing of *HSP* genes, thereby enabling *HSPs* to perform their corresponding biological functions, which means that they play an important role in plant resistance to stress [60]. In *Arabidopsis thaliana*, *HSFA1a*-overexpressed plants were found to increase the expression of *HSP70* and *HSP18.2* under adverse conditions, thus suggesting that *HSFA1a* has an inductive effect on the expression of *HSP70* and *HSP18.2*, (which, in turn, enhances plant resistance) [61]. In our study, *HSF1* (*HSFA1A*) and *HSP18.2* genes were significantly up-regulated in Comparisons A and C under the DIF. Therefore, we speculated that the expression of HSP18.2 may be regulated through HSF1, responding to the DIF to maintain normal plant growth and development.

## 4. Materials and Methods

### 4.1. Materials

The PLBs of *D. officinale* were provided by Horticultural Plant Biotechnology Institute at Fujian Agriculture and Forestry University. The reagents were purchased from Sigma-Aldrich (Sigma-Aldrich, St. Louis, MO, USA).

Culture flasks were inoculated with 1 g of stabilized PLBs. The masses of protocorms are yellow-green, with sizes around 10 × 10 cm. The medium consisted of 1/2 MS, 50 g/L mashed potato, 25 g/L sucrose, 6 g/L agar, and pH was adjusted to 5.4–5.8. After inoculation, cultures were incubated in a photobioreactor at 25/25 °C (day/night) for 30 d. Immediately after 30 d, DIF treatments were conducted in different photobioreactors (Table 3). The light intensity of the photobioreactors was 2000–2500 Lx, with a 12 h/d light duration. After 5 d, 10 bottles of PLBs were randomly selected from each treatment. The aggregates were mixed and weighed, and one part was used for the determination of the physiological and functional indexes, while another part was stored at −80 °C after quick freezing via liquid nitrogen for verification through qPCR after freezing. Three biological replicates were performed for each treatment comparison.

### 4.2. Determination of Polysaccharide and Flavonoid

The polysaccharide test was performed as described by Yang et al. [62] with a slight modification, i.e., the determination of the polysaccharide content in the PLBs was conducted using petroleum ether degreasing, and the Sevage method was utilized for protein removal.

The flavonoid content was measured according to Yuan’s method [1], and the absorbance of the reaction mixture at 510 nm was measured with a UV spectrophotometer. Rutin was used as a flavonoid standard curve.

### 4.3. Determination of Chlorophyll and Carotenoids

A mixed-liquid-extraction method was adopted for the determination, conducted in accordance with the previous report [63]. Briefly, the mixed extract was absolute (ethanol: acetone: distilled water = 4.5:4.5:1), and the OD values were obtained at the wavelength of 440 nm, 645 nm, and 663 nm. Finally, the contents of the chlorophyll a, chlorophyll b, carotenoids, chlorophyll a/b, and chlorophyll a + b were calculated using the equations described by Arnon [64], and three replicates were performed for each sample.
Chlorophyll a = (12.7 × OD663 − 2.69 × OD645) × V/1000 × M;
Chlorophyll b = (22.9 × OD645 − 4.68 × OD663) × V/1000 × M;
Carotenoids = (4.7 × OD440 − 0.27 × 20.21 × OD645 + 0.82 × OD663) × V/1000 × M.

### 4.4. Measurement of Alkaloid

After drying at 60 °C and grinding, the sample powders were screened using a 60-mesh sieve. The 0.10 g powder of the collected PLB sample was used to conduct alkaloid determination. Alkaloid extraction and dissolving were performed as described by Yuan et al. [1]. The absorbance was measured at a 416 nm wavelength. The total alkaloid content was measured using a dendrobine standard, and the concentration was expressed by alkaloid content (%) as per the following equation (samples were tested in triplicate).
Alkaloid content (%) = Dendrobine concentration (obtained by standard curve) × 50 (dilution factor) ÷ 400,000 (unit conversion) × 100%.

### 4.5. RNA Isolation, Library Preparation, and Sequencing

The PLBs of *D. officinale* after the DIF treatments (Table 3) were collected and frozen under liquid nitrogen. The samples were stored at −80 °C for further RNA exaction. Under the liquid nitrogen, 100 mg frozen samples were ground into a fine powder with a mortar and pestle. Three biological replicates were set up for each treatment. Total RNA was extracted by using BIOFLUX total RNA extraction kits (Hangzhou, China), following the manufacturer’s protocol. The quality and quantity of the PLBs’ RNA were determined using a NanoDrop 2000 (Thermo Fisher Scientific, Waltham, MA, USA).

The analysis of concentration, RNA integrity number (RIN), and value of the total RNA were determined with an Agilent 2100 Bioanalyzer (Agilent Technologies, Santa Clara, CA, USA). A TruSeq Stranded mRNA LT Sample Prep Kit (Illumina, San Diego, CA, USA) was used to prepare the library, and it was used in accordance with the manufacturer’s protocol. After purification, the libraries were validated by using an Agilent 2100 Bioanalyzer and an ABI StepOnePlus Real-Time PCR System. The qualified libraries were subjected to sequencing on an Illumina Hiseq 4000 platform, which was performed by the BGI Company (Shenzhen, China). For each sample, 6.67 Gb of raw data was generated with 150 bp/125 bp-length paired-end reads.

### 4.6. Transcriptome Mapping, Annotation, and Differential Expression Analysis

To obtain clean reads, the raw sequences were subjected to SOAPnuke software (v1.5.2, https://github.com/BGI-flexlab/SOAPnuke, accessed on date 11 March 2024) to remove the low-quality reads and unknown ploy-N (>5%). The clean reads were aligned with the reference map of *D. officinale* (https://www.ncbi.nlm.nih.gov/bioproject/262478, accessed on 15 September 2017) [65,66] on HISAT2 (Hierarchical Indexing for Spliced Alignment of Transcripts) software (Version v.2.0.4). Gene-structure extensions and novel-transcript annotations were conducted using certain tools, including StringTie [67], Trapnel [68], and Kong [69]. Based on the assembled map, low-quality SNPs and INDELs were identified and removed with GATK [70]. A differential splicing gene (DSG) was detected and filtered using rMATS [71] with an FDR threshold of ≤0.05. The clean reads were subjected to Bowtie2 [72] and RSEM [73] to calculate the level of gene expression. The PossionDis algorithm was applied to identify the DEGs with an FDR of ≤0.01 and a fold change of more than 2 times. Gene ontology (GO) annotations were used to characterize the gene functions, and Kyoto Encyclopedia of Genes and Genomes (KEGG) mapping was applied to identify the metabolic pathways. Gene preconcentration analysis was performed with the phyper function in R language, with an FDR of ≤0.01 representing significant preconcentration.

### 4.7. Synthesis Pathway Identification via RT-PCR Analysis

To understand the impact of the temperature on the synthetic pathway at a molecular level, the significantly differentially expressed genes involved in the pathways—including the starch sucrose synthesis pathway, pentose phosphate pathway, phenylpropane pathway, flavonol pathway, heat-stress defense, lignin-synthesis pathway, photosynthesis, and biological-clock pathway—were investigated in order to validate the expression difference, and this was achieved using an ABI StepOnePlus Real-Time polymerase chain reaction (PCR) system. The eighteen primers were designed in DNAMAN software (V6.0), as listed in Table 2. The 18s rRNA was used as a housekeeping gene [74,75,76]. Transcription was conducted using TransScript^®^ miRNA First-Strand cDNA Synthesis SuperMix (AT311, TransGen Biotech, Beijing, China) following the manufacturer’s instructions. A PCR-amplification system at 94 °C for 30s was followed by 45 cycles of 94 °C for 5 s and 60 °C for 30 s. The relative gene expression was calculated and analyzed with the 2^−△△CT^ method.

### 4.8. Statistical Analysis

Three biological replicates were set for each treatment, and SPSS Statistics 17.0 software was used to analyze the experimental data. The analysis of variance was performed with Duncan’s method, and the significance level was *p* < 0.05.

## 5. Conclusions

In conclusion, the transcriptome analysis indicated that DEGs respond to temperature changes in various ways through differential expression. The analysis of these DEGs provides a reference for studying the growth and development of plants, as well as the accumulation of functional metabolites under a DIF.

## Figures and Tables

**Figure 1 plants-13-00874-f001:**
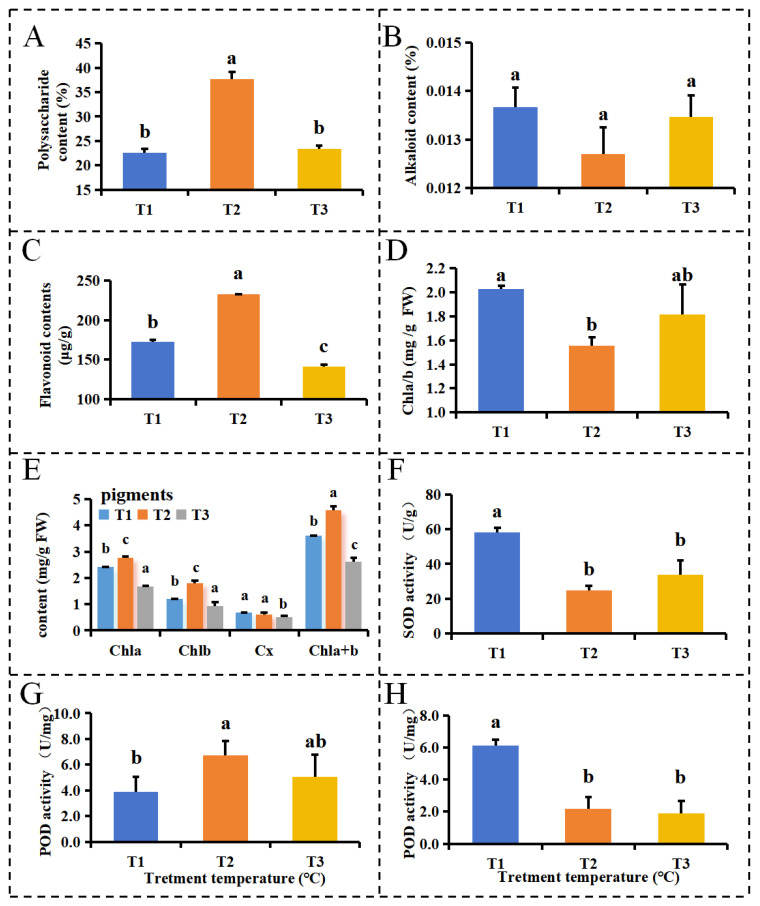
Metabolism contents of PLBs under the different temperature treatments. Polysaccharide contents (**A**), alkaloid contents (**B**), flavonoid contents (**C**), chlorophyll a/b (**D**), pigment content (**E**), SOD activity (**F**), POD activity (**G**), and MDA content (**H**). T1:13/13 °C; T2:25/13 °C; T3: 25/25 °C. For each treatment, three biological repeats were performed. Results are presented as mean values of three replicates. Different lower-case letters indicate significant differences. *p* < 0.05.

**Figure 2 plants-13-00874-f002:**
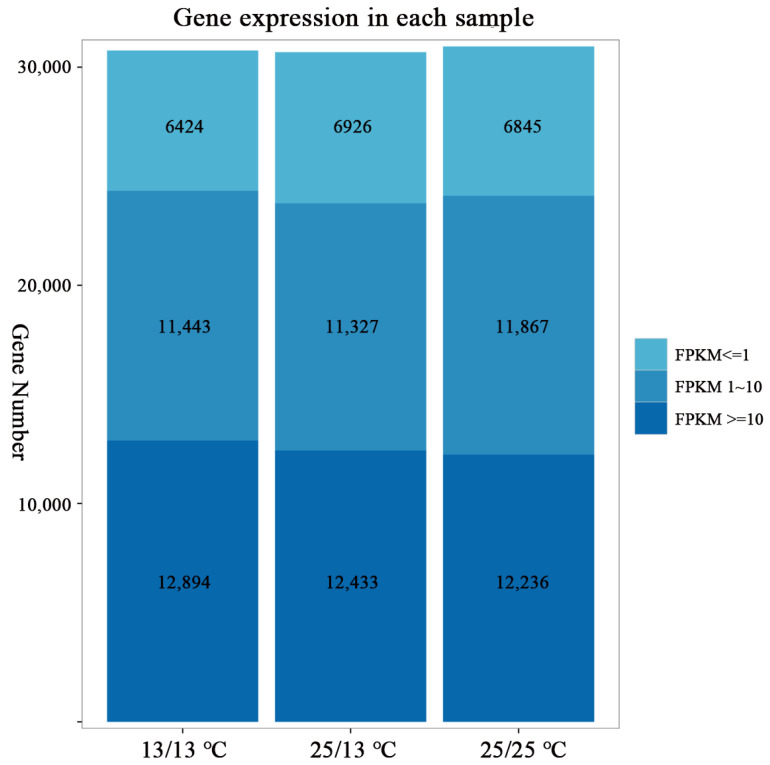
Gene expression in each sample.

**Figure 3 plants-13-00874-f003:**
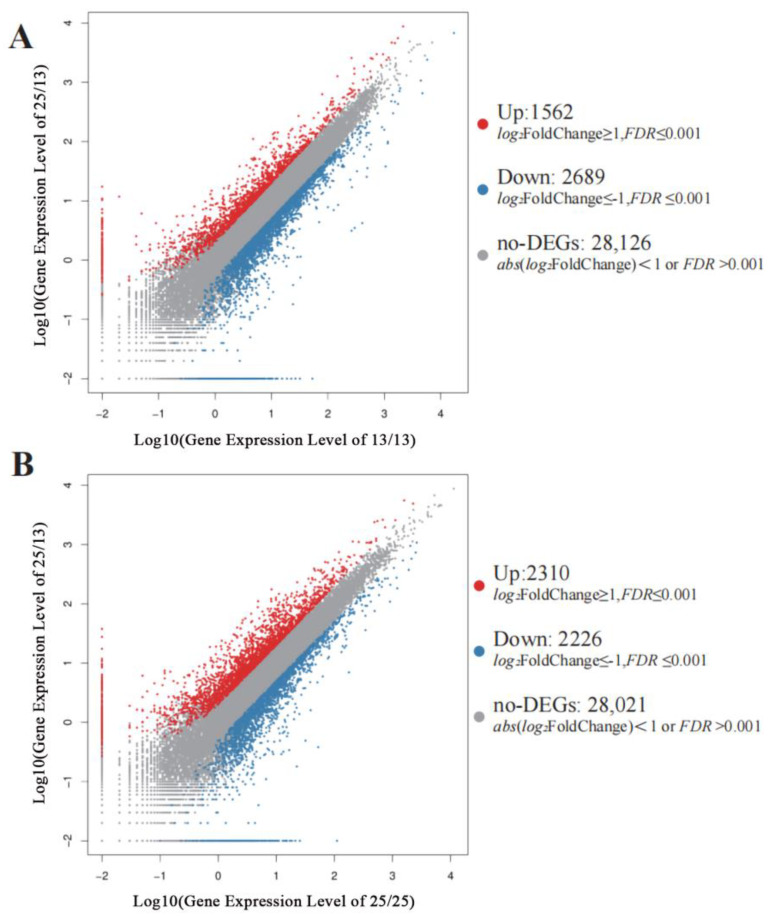
MA-plot analysis of DEGs in *D. officinale* PLBs. Up-DEGs are shown in red; down-DEGs are shown in blue; no-DEGs are shown in gray. The *X* and *Y* axes represent the logarithmic values of gene expression. (**A**) Comparison A (25/13 °C vs. 13/13 °C); (**B**) Comparison C (25/13 °C vs. 25/25 °C).

**Figure 4 plants-13-00874-f004:**
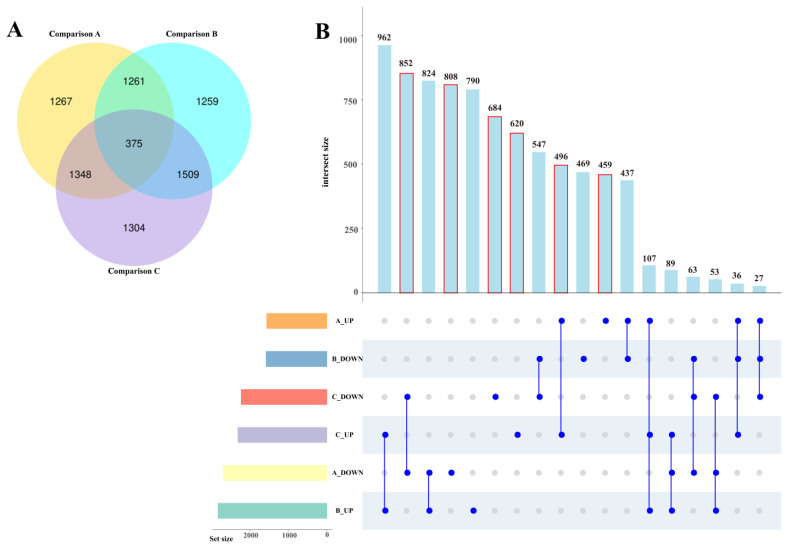
Numbers of DEGs in different comparisons; (**A**) Venn diagram of DEGs; (**B**) Numbers of up-/down-regulated DEGs in different Comparisons; A_UP is the up-regulated DEGs in Comparison A, B_DOWN is the down-regulated DEGs in Comparison B, C_DOWN is the down-regulated DEGs in Comparison C, C_UP is the up-regulated DEGs in Comparison C, A_DOWN is the down-regulated DEGs in Comparison A, B_UP is the up-regulated DEGs in Comparison B. The red box shows the numbers of DIF-specific expressed DEGs.

**Figure 5 plants-13-00874-f005:**
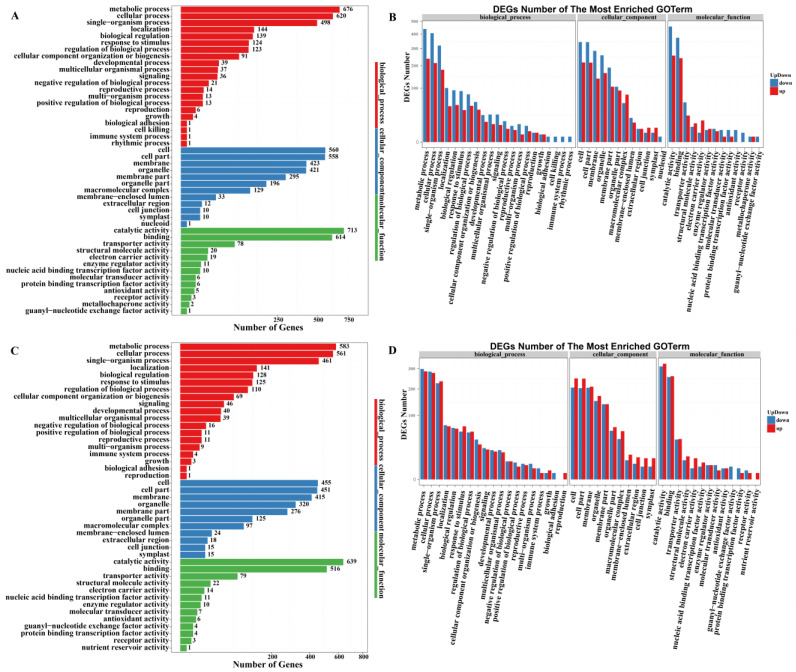
GO functional classification of DEGs. (**A**,**C**) The number of DEGs; *X*-axis: the number of DEGs; *Y*-axis: the GO functional classifications; the biological process is shown in red; the cellular component is shown in blue; the molecular function is shown in green. (**B**,**D**) the number of up-/down-regulated DEGs; *X*-axis: the GO functional classifications; *Y*-axis: the number of DEGs; up-regulated DEGs is shown in red; and the down-regulated DEGs is shown in blue. (**A**,**B**) Comparison A (25/13 °C vs. 13/13 °C); (**C**,**D**) Comparison C (25/13 °C vs. 25/25 °C); DEGs (log_2_Fold(change) ≥ 1 or FDR ≤ 0.01).

**Figure 6 plants-13-00874-f006:**
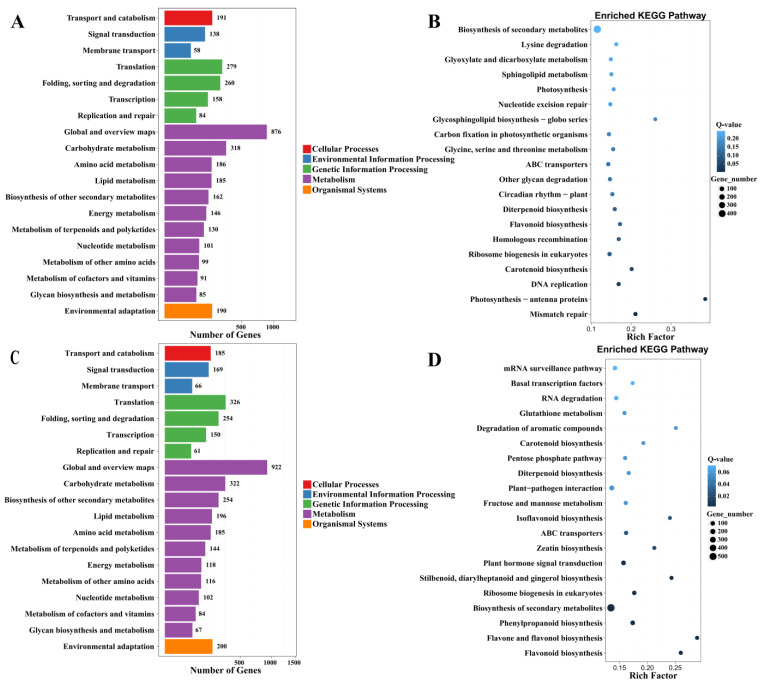
KEGG pathway classification of significantly enriched DEGs. (**A**,**C**) The pathway classification of DEGs; *X*-axis: the number of DEGs; *Y*-axis: the functional classification of KEGG; cellular processes are shown in red; environmental information processing is shown in blue; genetic information processing is shown in green; metabolism is shown in purple; organismal systems are shown in orange. (**B**,**D**) Results of pathway enrichment of differentially expressed genes; *X*-axis: the rich factor; *Y*-axis: the KEGG pathways; Q-values are shown in blue. High q-values are shown in light blue. Low q-values are shown in dark blue. The lower the q-value, the more significant the enrichment result is. The size of the points represents the number of DEGs. The bigger the point’s size, the higher the DEGs’ number. (**A**,**B**) Comparison A (25/13 °C vs. 13/13 °C); (**C**,**D**) Comparison C (25/13 °C vs. 25/25 °C). DEGs (log_2_Fold(change) ≥ 1 or FDR ≤ 0.01).

**Figure 7 plants-13-00874-f007:**
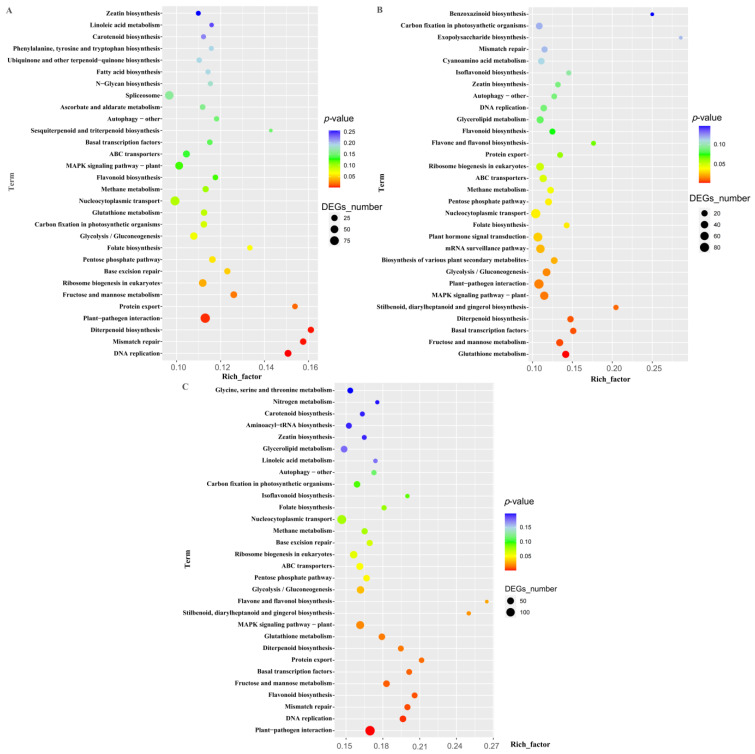
KEGG analysis of the DEGs that specifically respond to DIF, (**A**) Comparison A filtered out DEGs from Comparison B, (**B**) Comparison C filtered out DEGs from Comparison B, (**C**) Comparison A and Comparison C filtered out DEGs from Comparison B.

**Figure 8 plants-13-00874-f008:**
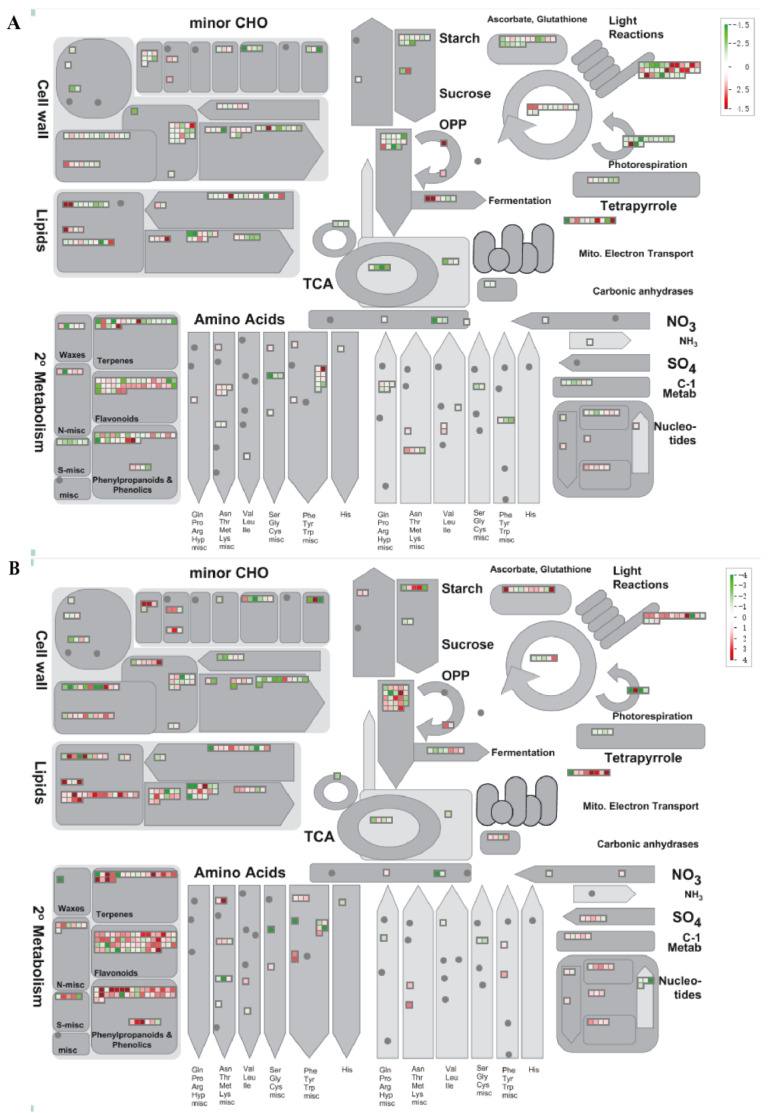
Metabolism overview of *D. officinale* PLBs under different temperatures. (**A**) Comparison A (25/13 °C vs. 13/13 °C); (**B**) Comparison C (25/13 °C vs. 25/25 °C). Genes that were shown to be differentially expressed using *p* < 0.05 as a cutoff value were imported. Each square represents a DEG. Red represents up-regulated DEGs. Green represents down-regulated DEGs. The color density represents the expression level of DEGs.

**Figure 9 plants-13-00874-f009:**
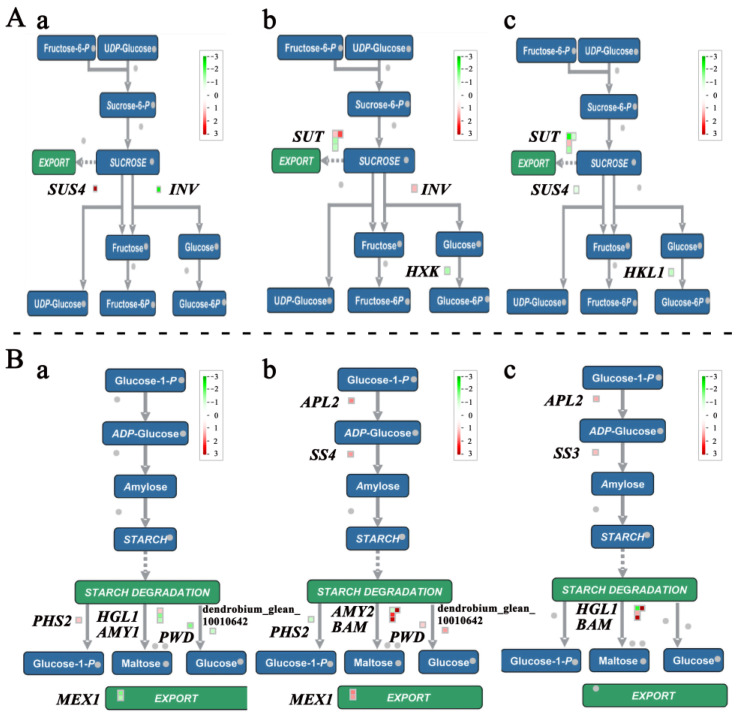
DEGs related to polysaccharide metabolism. (**A**) Effects of DIF on sucrose synthesis and metabolism-related genes of *D. officinale* PLBs. Each square represents a DEG. Red represents up- regulated DEGs. Green represents down-regulated DEGs. The color density represents the expression level of DEGs. (**B**) Effects of DIF on the starch synthesis- and metabolism-related genes of *D. officinale* PLBs. Genes that were shown to be differentially expressed using *p* < 0.01 as a cutoff value were imported. From left to right, a–c are denoted as Comparisons A–C. a: Comparison A (25/13 °C vs. 13/13 °C); b: Comparison B (13/13 °C vs. 25/25 °C); c: Comparison C (25/13 °C vs. 25/25 °C).

**Figure 10 plants-13-00874-f010:**
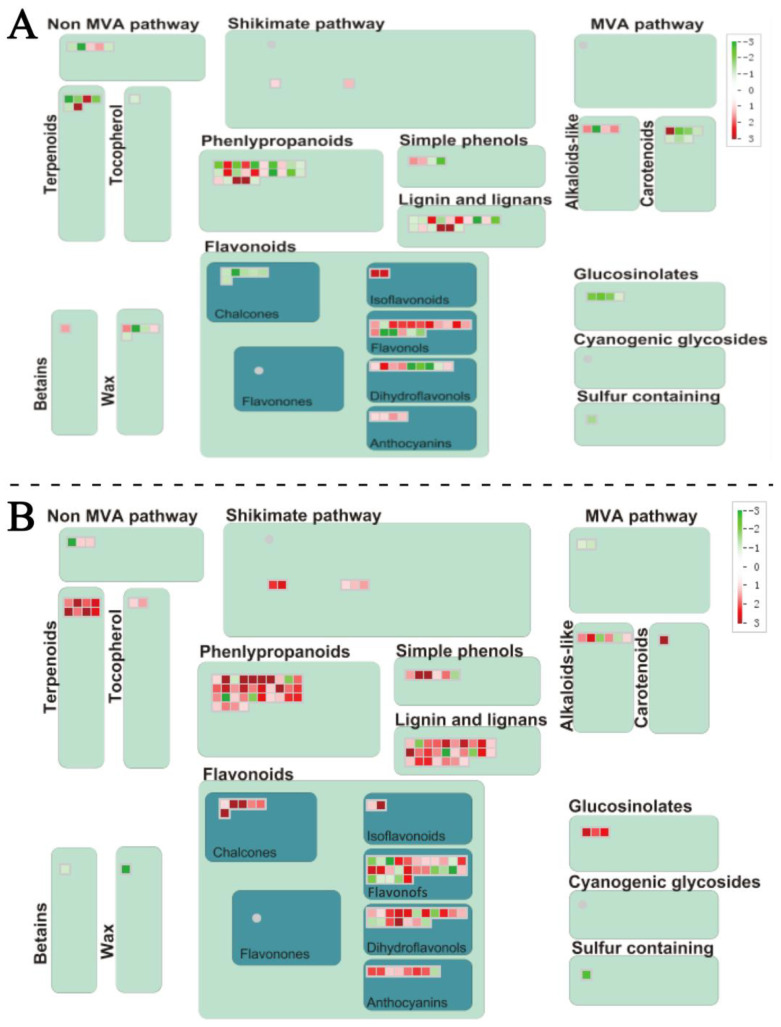
Secondary metabolic pathways of *D. officinale* PLBs. Each square represents a DEG. Red represents up-regulated DEGs. Green represents down-regulated DEGs. The color density represents the expression level of DEGs. (**A**) Comparison A (25/13 °C vs. 13/13 °C); (**B**) Comparison C (25/13 °C vs. 25/25 °C). Genes that were shown to be differentially expressed using *p* < 0.01 as a cutoff value were imported.

**Figure 11 plants-13-00874-f011:**
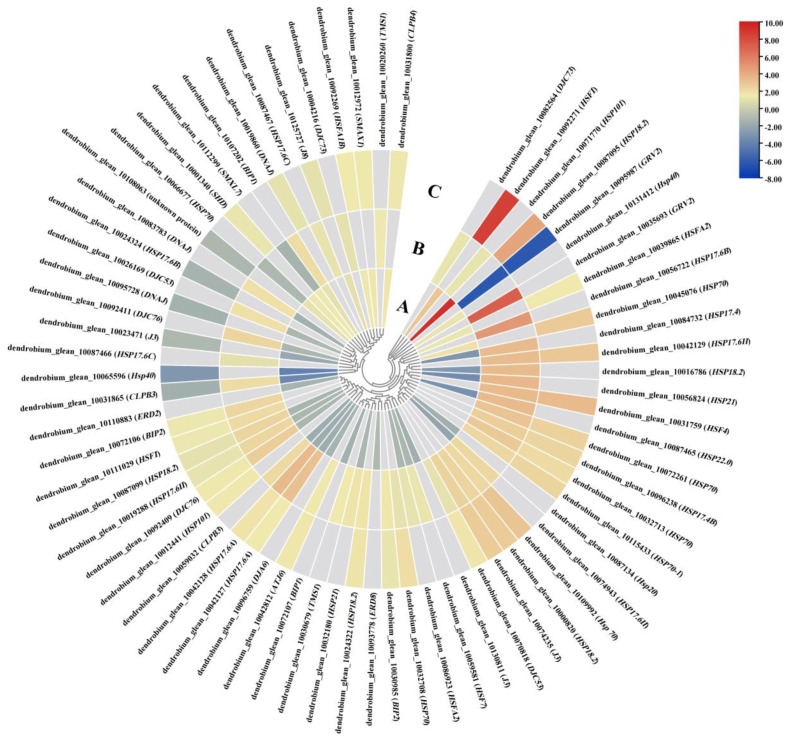
The expression analysis of DEGs in the heat-stress pathway. A: Comparison A (25/13 °C vs. 13/13 °C); B: Comparison B (25/25 °C vs. 13/13 °C); C: Comparison C (25/13 °C vs. 25/25 °C). Genes that were shown to be differentially expressed using *p* < 0.01 as a cutoff value were imported.

**Figure 12 plants-13-00874-f012:**
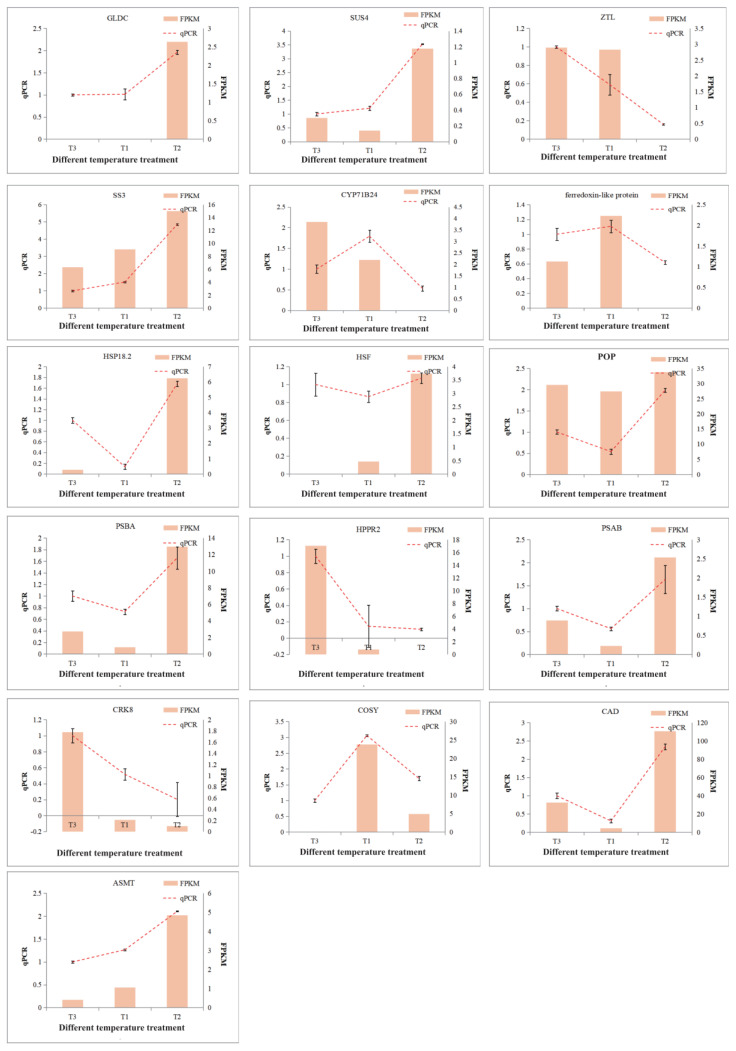
QRT-PCR analysis of candidate genes for temperature response in *D. officinale* PLBs. The column chart shows the FPKM value. The line graph shows the relative expression of genes.

**Table 1 plants-13-00874-t001:** Summary of the sequencing data in each sample.

Sample	T1 (13/13 °C)	T2 (25/13 °C)	T3 (25/25 °C)
Total Raw Reads (Mb)	69.97	69.97	69.97
Total Clean Reads (Mb)	66.98	66.54	66.66
Total Clean Bases (Gb)	6.7	6.65	6.67
Clean Reads Q20 (%)	99.38	99.35	99.36
Clean Reads Q30 (%)	97.88	97.77	97.8
Clean Reads Ratio (%)	95.73	95.11	95.27
Total Mapping Ratio (%)	82.16	82.40	81.39
Uniquely Mapping Ratio (%)	54.89	54.24	53.10

**Table 2 plants-13-00874-t002:** Primers used for real-time PCR analysis of *D. officinale* PLBs.

NO	Name	Primers	Sequences (5′–3′)
1	*CRK8*	dendrobium_glean_10017232-F	TATCGCCTCATGCTTCCACTA
		dendrobium_glean_10017232-R	GACACTCTCTCCGCTCTCTT
2	*CAD6*	dendrobium_glean_10107234-F	CTATGGGGATGACTGCCACT
		dendrobium_glean_10107234-R	CTACCGATTGCCTCCTTCAG
3	*COSY*	dendrobium_glean_10128767-F	ACTACGAAAAGGGCATCACG
		dendrobium_glean_10128767-R	CTCACCTTCCGATTCAGCAT
4	*CYP71B24*	dendrobium_glean_10021134-F	CCAGAGTTGCGATGGAAGAT
		dendrobium_glean_10021134-R	TGGCTTATCCCAGTTTTTGG
5	*ASMT*	dendrobium_glean_10048936-F	GCGAGTGCTACATGACTGGA
		dendrobium_glean_10048936-R	CTGTCCTCTCTTTCCCACCA
6	*PSAB*	dendrobium_glean_10008093-F	TACAATCGGATTACGCACCA
		dendrobium_glean_10008093-R	AACGCTTGGTTTCCATTTTG
7	*PSBA*	dendrobium_glean_10135332-F	GGGTCGTGAGTGGGAACTTA
		dendrobium_glean_10135332-R	TGTGCTCTGCCTGGAATACA
8	*CHS*	dendrobium_glean_10103920-F	TACTGAACGACCGCTTTTCC
		dendrobium_glean_10103920-R	TGCCTCCACGAGACTCTTTT
9	*HSF1*	dendrobium_glean_10092271-F	AAACGGCAAGGACAACAAAG
		dendrobium_glean_10092271-R	GCAGGATGTGAACGAAACCT
10	*HPPR2*	dendrobium_glean_10101025-F	TGTGTATTGATCGCCTCTGC
		dendrobium_glean_10101025-R	AAGACCAATGATGCCTACCG
11	*PFK3*	dendrobium_glean_10138387-F	GCAAGCAGAGATGTGGACTG
		dendrobium_glean_10138387-R	CACCCTCAGCAACAACGATA
12	POP	dendrobium_glean_10121871-F	CCGCCAGAGATGTGTTGTAG
		dendrobium_glean_10121871-R	CTGTAACCAGGCTGACTGAATC
13	*HSP18.2*	dendrobium_glean_10087095-F	CGATTCCGACTTCCTGAGAA
		dendrobium_glean_10087095-R	GGCTTCTTGACCTCCTCCTT
14	*ZTL*	dendrobium_glean_10106356-F	CCCAACAGAGGAGAAACCAA
		dendrobium_glean_10106356-R	GCAATTCGCTTAGCATCCAT
15	*SUS4*	dendrobium_glean_10105017-F	GGTGGTCCAGCAGAGATCAT
		dendrobium_glean_10105017-R	CCAGCCAAGGTCATCAATCT
16	*SS3*	dendrobium_glean_10016137-F	GGATTTCCATGCCGCTATAA
		dendrobium_glean_10016137-R	CAAACGTGCTGCTTTCTCAG
17	*GLDC*	dendrobium_glean_10031276-F	CTCTGGGCTTCTTCAACAGC
		dendrobium_glean_10031276-R	TGAGCATGGTCAGTGATTCC
18	ferredoxin-like protein	dendrobium_glean_10136528-F	TGGTGTCCTAGCCCAGACTC
		dendrobium_glean_10136528-R	CCGACCTCTGTGTGGTTTTT

**Table 3 plants-13-00874-t003:** DIF-processing temperature setting.

Treatment	DIF Treatment Phase (5 Days)		
	T1	T2	T3 (Control)
8:00~20:00	13 °C	25 °C	25 °C
20:00~8:00	13 °C	13 °C	25 °C

## Data Availability

The data presented in this study are available on request from the corresponding author due to (the relevant papers have not yet been published). All the data covered in the manuscript are included in the Appendix A.

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
