# Peer review of "Transcriptomic Analysis for Diurnal Temperature Differences Reveals Gene-Regulation-Network Response to Accumulation of Bioactive Ingredients of Protocorm-like Bodies in Dendrobium officinale"

_plants, 2024, doi:10.3390/plants13060874_

Round 1

Reviewer 1 Report

Comments and Suggestions for Authors

The manuscript of Chen et al. presents a transcriptomic analysis of protocorm-like bodies (PLBs) of Dendrobium officinale grown in vitro in three different temperature conditions. They aimed to investigate the effect of day/night temperature variations treatment on functional ingredients in D. officinale PLBs. D. officinale contains various bioactive ingredients, including polysaccharides and alkaloids. After the growth of PLBs in constant temperature (25/25°C; day/night) for 30 days (pretreatment), they used three temperature combinations (13/13°C – T1, 25/13°C – T2, and 25/25°C – T3) for 5 days. Samples were collected for RNA extraction, RNA-Seq analysis, and differentially expressed genes (DEGs) identification. Some DEGs were used for RT-qPCR and comparison with the RNA-Seq results. Additionally, the authors analyzed functional metabolites in D. officinale PLBs. The study is interesting, and the authors generated a lot of data. However, the results focused mainly on numbers of the transcriptomic data (GO enrichment, KEGG classification, etc.). Moreover, the authors do not make their conclusion clear considering what is ideal regarding PLBs’ composition and growth conditions. They obtained more polysaccharides and more flavonoids in T2. However, PLBs grown at T2 seem to have fewer alkaloids. Is T2 the best condition for PLB culture? What is more important: high diurnal temperature OR day/night temperature variation?

Below is a list of points that need attention from the authors. These points do not exhaust all the issues found but are suggestions for improvement.

Lines 11-13: Please correct the day temperature of T2 (25/13°C). In several places of the manuscript, it is written the opposite (13/25°C), as here. Please also rewrite the experimental condition to make clear that 25/25°C for 30 days was a pretreatment.

Lines 15-16: Please revise the English. Along the manuscript, there are some issues as here: “… the associated genes WERE analyzed by qPCR. The RESULTS showed…” Please also correct the word TREATMENT in several graphics of Figure 1.

Lines 93-98: Please define T1, T2, and T3 before starting to present the results obtained. This information only appears later in the manuscript.

Lines 108-109: “… treatment for 5 days had no significant effect on the growth amount. We measured photosynthetic pigments and antioxidant enzyme activities…”. Would it not be better to evaluate biomass to study growth?

Throughout the manuscript, the authors use abbreviations, such as MDA values (line 122), before introducing them to the readers. They also refer to the names of several genes without explaining their names and/or functions.

Line 144: Please give a reference for the “PossionDis method”.

Lines 144-149: Is the genome of D. officinale available? What is the meaning of these numbers of DEGs and NO-DEGS?

Line 562: How many biological samples were used for RT-qPCR analysis?

Lines 569-570: I believe 18S rRNA is inappropriate for this experiment as an endogenous gene control.

Please provide detailed legends for each Figure. In the present form, they are not very informative.

Comments on the Quality of English Language

As mentioned in the comments for the authors, it is important to revise the English. Along the manuscript, there are some issues as here: “… the associated genes WERE analyzed by qPCR. The RESULTS showed…” (lines 15-16). 

Author Response

The itemized list of changes addressing reviewer comments

Dear respected editor and reviewers,

Thank you so much for your great comments on our manuscript entitled ‘Transcriptomic analysis for diurnal temperature differences reveals gene regulation network response to accumulation of bioactive ingredients of protocorm-like bodies in Dendrobium officinale’, which was submitted to the Plants (Basel) (Manuscript ID: plants-2816909). Your comments helped us a lot to improve its quality. Based on your comments, we had corrected our manuscripts carefully. Now we resubmit the revised manuscript to your journal and hope it would be considered for publication. Responses to each comment were listed as follows:

Response to Reviewer 1:

The study is interesting, and the authors generated a lot of data. However, the results focused mainly on numbers of the transcriptomic data (GO enrichment, KEGG classification, etc.). Moreover, the authors do not make their conclusion clear considering what is ideal regarding PLBs’ composition and growth conditions. They obtained more polysaccharides and more flavonoids in T2. However, PLBs grown at T2 seem to have fewer alkaloids. Is T2 the best condition for PLB culture? What is more important: high diurnal temperature OR day/night temperature variation?

Reply: Thank you for your professional comments. In this version of revised manuscript, we added the discussion of the effect of temperature difference on functional products including polysaccharides, flavonoids and alkaloids (Lines 434-468, Page19-20) . Since polysaccharide has been proved to be the most important functional product of D.officinale, which is an indicator for evaluating the quality of D.officinale, we used the polysaccharide content as the most important indicator of functional product. [Xu J,Li S L,Yue R Q,et al. A novel and rapid HPGPC-based strategy for quality control of saccharide-dominant herbal materials:Dendrobium officinale, a case study[J].Analytical & Bioanalytical Chemistry,2014,406(25):6409-6417.DOI:10.1007/s00216-014-8060-9.]The alkaloid content had no changed significantly under temperature difference (T2: 25/13℃) than that under constant temperature (T1: 13/13℃and T3: 25/25℃). Therefore, we concluded that the temperature difference was more favourable to promote the accumulation of functional metabolites of D.officinale PLBs compared with the constant temperature treatment. 

Comment 1: Lines 11-13: Please correct the day temperature of T2 (25/13°C). In several places of the manuscript, it is written the opposite (13/25°C), as here. Please also rewrite the experimental condition to make clear that 25/25°C for 30 days was a pretreatment.

Reply: Thank you for pointing out these problems for us. We have corrected the errors in the revised manuscript (Line13, Page 1). And according to your comments, we have rewrote the experimental condition, 25/25°C for 30 days was not a pretreatment, but a the culture stage of the PLBs. (Lines 575-586, Page 22). 

Comment 2: Lines 15-16: Please revise the English. Along the manuscript, there are some issues as here: “… the associated genes WERE analyzed by qPCR. The RESULTS showed…” Please also correct the word TREATMENT in several graphics of Figure 1.

Reply: Thank you for your great comments. We had made modifications in the revised manuscript. And the last revised version of the manuscript had been edited by native English-speakers(MDPI  Rrecommending editing service-AUTHORservices to polish the language.

Comment 3: Lines 93-98: Please define T1, T2, and T3 before starting to present the results obtained. This information only appears later in the manuscript.

Reply: Thank you for pointing out these problems for us. We had added the definitions of T1, T2, and T3 in the manuscript when they appeared for the first time (Line 81, Page 2).

Comment 4: Lines 108-109: “… treatment for 5 days had no significant effect on the growth amount. We measured photosynthetic pigments and antioxidant enzyme activities…”. Would it not be better to evaluate biomass to study growth?

Reply: Thank you for your professional comments. The D.officinale PLBs grew slowly and thus had very little growth for 5 d. In our study, we mainly focused on the effect of DIF on the accumulation of functional metabolites. Therefore, we used these physiological indicators to assess the physiological changes of DIF on D.officinale PLBs. In our revised manuscript, we had modified the title "2.2 Analysis of the effect of DIF on physiological indices of D.officinale PLBs" to better accurately aligned with our objective (Line 101, Page 3).

Comment 5: Throughout the manuscript, the authors use abbreviations, such as MDA values (line 122), before introducing them to the readers. They also refer to the names of several genes without explaining their names and/or functions.

Reply: Thank you for pointing out these problems for us. We had supplemented the introductions to acronyms, and the full names of genes in our revised manuscript.

Comment 6: Line 144: Please give a reference for the ossi“PonDis method”.

Reply: we had added the reference for the “PossionDis method” in the revised manuscript (Line 145, Page 5).

Comment 7: Lines 144-149: Is the genome of D. officinale available? What is the meaning of these numbers of DEGs and NO-DEGS?

Reply: In previous studies, it was found that Dendrobium officinale(D. officinale) and Dendrobium catenatum(D. catenatum) are synonyms.[Jin-Ping, S.I., et al., Herbal textual research on relationship between Chinese medicine"Shihu"(Dendrobium spp.) and "Tiepi Shihu"(D. catenatum). China Journal of Chinese Materia Medica, 2017. 42(10): p. 2001-2005.] A draft genome sequence of Dendrobium officinale Kimura & Migo has been reported in 2014, but the highly fragmented assembly and the presence of multiple peaks in K-mer analyses, suggesting that its sequence was likely derived from an artificial hybrid, seriously complicate correct interpretation of the genome. The genome sequence we used was D. catenatum, which had been reported in 2016, and the genome assembly of D.officinale is of high quality.[Zhang GQ, Xu Q, Bian C, et al. The Dendrobium catenatum Lindl. genome sequence provides insights into polysaccharide synthase, floral development and adaptive evolution. Sci Rep. 2016 Jan 12;6:19029. doi: 10.1038/srep19029.] “A total of 222.51 Gb of raw reads were generated, with multiple insert libraries ranging in size from 180 bp to 20 Kb. A K-mer analysis estimated the genome size of D. catenatum at 1.11 Gb. Mapping all of the paired-end reads to the assembly revealed that 97% of the sequence had a coverage depth greater than five. Further quality analysis indicated that 93% of the set of eukaryotic core genes (CEGMA) 10 were present and 97% were partially represented, suggesting near completeness of the euchromatin component. In addition, 93%–95% of the RNA seq data set could be mapped onto the assembled sequence. These results suggest that our genome assembly is of high quality.” 

DEGs indicates the differentially expressed genes. Gene expression was compared under three different temperature treatments. Differentially expressed genes were defined by default as genes with FDR ≤ 0.01, and log2Fold change≥1 or ≤ -1. The NO-DEGs indicates the genes which expression were not significantly changed among different treatment comparisons. We had added the descriptions on Lines147-154, Page 5.

Comment 8: Line 562: How many biological samples were used for RT-qPCR analysis?

Reply: Three biological replicates were performed for each treatment comparison. We had added the descriptions on Lines 575-586, Page 22; Lines 622-625, Page 23.

Comment 9: Lines 569-570: I believe 18S rRNA is inappropriate for this experiment as an endogenous gene control.

Reply: We searched the relevant literature, and the 18S rRNA was commonly used as reference gene for qRT-PCR of D. officinale. The Ct values of this gene were stable in different treatments in our experiment. So, we think 18S rRNA is appropriate for this experiment as an endogenous gene control. In our revised manuscript, we had added the citations (Line 663, Page 23).[[1]Zhang, G., et al., Reference Gene Selection for Real-Time Quantitative PCR Analysis of Dendrobium officinale. Journal of Chinese Pharmaceutical ences, 2013. 48(19): p. 1664-1668. [2] Ling, H., X. Zeng and S. Guo, Functional insights into the late embryogenesis abundant (LEA) protein family from Dendrobium officinale (Orchidaceae) using an Escherichia coli system. Scientific Reports, 2016. 6(1): p. 39693. [3] Wang, et al., Molecular characterization and expression analysis of WRKY family genes in Dendrobium officinale. Genes & Genomics, 2018 (40):265–279.]

Comment 10: Please provide detailed legends for each Figure. In the present form, they are not very informative.

Reply: Thank you for pointing out these problems for us. In our revised manuscript, we had made the modifications accordingly.

Reviewer 2 Report

Comments and Suggestions for Authors

In this article, authors used transcriptome analysis on a traditional medicine herb. My comments are as follows:

1. title: How come only ‘Temperature’ and ‘Protocorm’ are capitalized?

2. Page 1, lines 7-9: Please provide affiliations.

3. Page 1, line 16: Does ‘Diurnal’ have to be capitalized?

4. Page 1, line 41: D. officinale should be italic.

5. Page 2, line 65, 68, 69, and 71: Why ‘Temperature’ is capitalized?

6. Page 2: Line 73 and 76: Dendrobium officinale should be italic.

7. Page 3: lines 99-102 and lines 103-105: One sentence as one paragraph. A mind suggestion, the authors can combine them into one paragraph.

8. Page 3, line 111: (Chla) is a type and should be replaced by Chlb.

9. Page 3, line 116: SOD and POD are abbreviations. It is better to give the full names at first appear instead of giving the full name at page 16, line 369.

10. Page 7, Figure 2: Please enlarge the photos and increase resolution.

11. Page 15, Figure 7: Subfigures are too small, please improve.

12. Page 18, line 469: Dendrobium ironwood should be italic.

13. Page 18, line 490: Please provide city name for Sigma.

14. Page 18, line 503: (Yang et al., 2015b) should be replaced by [number].

Author Response

The itemized list of changes addressing reviewer comments

Dear respected editor and reviewers,

Thank you so much for your great comments on our manuscript entitled ‘Transcriptomic analysis for diurnal temperature differences reveals gene regulation network response to accumulation of bioactive ingredients of protocorm-like bodies in Dendrobium officinale’, which was submitted to the Plants (Basel) (Manuscript ID: plants-2816909). Your comments helped us a lot to improve its quality. Based on your comments, we had corrected our manuscripts carefully. Now we resubmit the revised manuscript to your journal and hope it would be considered for publication. Responses to each comment were listed as follows:

Response to Reviewer 2:

Comment 1: title: How come only ‘Temperature’ and ‘Protocorm’ are capitalized?

Reply: Thank you for pointing out this problem for us. We had corrected the errors accordingly(Lines 2- 4, Page 1).

Comment 2:Page 1, lines 7-9: Please provide affiliations.

Reply: We had added the affiliations in this version of revised manuscript (Lines 7, Page 1).

Comment 3:line 16: Does ‘Diurnal’ have to be capitalized?

Reply: Thank you for pointing out this problem for us. We had corrected the errors accordingly(Line 13, Page 1).

Comment 4:Page 1, line 41: D. officinale should be italic.

Reply: Thank you for pointing out this problem for us. We had corrected the errors accordingly(Line 35, Page 1).

Comment 5:Page 2, lines 65, 68, 69, and 71: Why ‘Temperature’ is capitalized?

Reply: Thank you for pointing out these problems for us. We had corrected the errors accordingly(Lines 61, 65, 66, and 67 Page 2).

Comment 6:Page 2: Lines 73 and 76: Dendrobium officinale should be italic.

Reply: Thank you for pointing out these problems for us. We had corrected the errors accordingly(Lines 69,72, Page 2).

Comment 7: Page 3: lines 99-102 and lines 103-105: One sentence as one paragraph. A mind suggestion, the authors can combine them into one paragraph.

Reply: Thank you for your great comments. We had combined them into one paragraph in this version of revised manuscript (Lines 93-100, Page 3).

Comment 8: Page 3, line 111: (Chla) is a type and should be replaced by Chlb.

Reply: Thank you for pointing out this problem for us. We had corrected the errors accordingly(Line 107, Page 3).

Comment 9: Page 3, line 116: SOD and POD are abbreviations. It is better to give the full names at first appear instead of giving the full name at page 16, line 369.

Reply: Thank you for pointing out these problems for us. We had corrected the errors accordingly(Line 113, Page 3).

Comment 10: Page 7, Figure 2: Please enlarge the photos and increase resolution.

Reply: We had replaced the figure accordingly.In order to make the image clearer, Figure 2 has been divided into three figures: Figure 2, Figure 3, and Figure 4. “Figure 2 MA-plot analysis of DEGs in D. officinale PLBs”; “Figure 3 GO functional classification of DEGs. (A,C,E): the number of DEGs. (B,D,F): the number of up-/down-regulated DEGs.”; “Figure 4 KEGG pathway classification of significantly enriched DEGs. (A,C,E): the pathway classification of DEGs; (B,D,F): Results of Pathway enrichment of differentially expressed genes.”

Comment 11: Page 15, Figure 7: Subfigures are too small, please improve.

Reply: Thank you for your great comments. We had made modifications accordingly(Lines 386-389, Figure 7, Page 17-18).

Comment 12: Page 18, line 469: Dendrobium ironwood should be italic.

Reply: Thank you for pointing out this problem for us. We had corrected the errors accordingly(Line 553, Page 22).

Comment 13:Page 18, line 490: Please provide city name for Sigma.

Reply: We had added the description accordingly(Line 574, Page 22).

Comment 14:Page 18, line 503: (Yang et al., 2015b) should be replaced by [number].

Reply: Thank you for pointing out this problem for us. We had corrected the errors accordingly (Line 593, Page 22).

Reviewer 3 Report

Comments and Suggestions for Authors

The authors test the effect of diurnal temperature variation on transcriptional regulation.  The data can be organized much better as the current version feels like authors are throwing lots of data without summarizing key points.  It would be helpful if authors can pull out key information to the potential readers.  In addition, some of the conclusions or authors’ claims are not well justified.   

Major comments

1.       Please, check author names and their affiliations

2.       Some of very technical details are included in the abstract, which is fine, but this seems to make the abstract unnecessarily long and tedious.  I suggest making the abstract more concise and summarize the results better.  For example, lines 14-17 can be shortened from “Then Illumina.. . The results showed” into “The analysis on transcriptome showed”.  Authors should move the technical information into the methods and focus on showcase their results in the main text.

3.       Line 31-37: Authors can hypothesize, but they need to show concrete evidences to show their  hypothesis is validated or disproved.   

4.       Figure 1.  What are the sample sizes?  Also, please, state the sample collection detail for example “immediately after 5 day treatment”

5.       Line 130-132: Which cells were used to for RNA-seq?

6.       Figure 2B/C: A more informative way to present this data would be to show the enrichment factor.  For example, 20 DEGs found in your comparison would be significant if 10 DEGs are expected, which means 2-factor enrichment.

7.       Lines 306-321: The role of HSP family proteins are not clear from this study.  The expression of those genes are changing, which does not mean DIF regulation occurs through them.  These genes might the results of temperature variations, not regulators of this process.

Minor comments

1.       Line 19-23: Move into the main text.

2.       Line 16: What is DIF?

Author Response

The itemized list of changes addressing reviewer comments

Dear respected editor and reviewers,

Thank you so much for your great comments on our manuscript entitled ‘Transcriptomic analysis for diurnal temperature differences reveals gene regulation network response to accumulation of bioactive ingredients of protocorm-like bodies in Dendrobium officinale’, which was submitted to the Plants (Basel) (Manuscript ID: plants-2816909). Your comments helped us a lot to improve its quality. Based on your comments, we had corrected our manuscripts carefully. Now we resubmit the revised manuscript to your journal and hope it would be considered for publication. Responses to each comment were listed as follows:

Response to Reviewer 3:

Comment 1:Please, check author names and their affiliations

Reply: Thank you for your comments. We had corrected the descriptions in our revised manuscript (Lines 7-10, Page 1).

Comment 2:Some of very technical details are included in the abstract, which is fine, but this seems to make the abstract unnecessarily long and tedious.  I suggest making the abstract more concise and summarize the results better.  For example, lines 14-17 can be shortened from “Then Illumina.. . The results showed” into “The analysis on transcriptome showed”.  Authors should move the technical information into the methods and focus on showcase their results in the main text.

Reply: Thank you for your great comments. We had removed technical details and made modifications in the abstract in our revised manuscript.

Comment 3: Line 31-37: Authors can hypothesize, but they need to show concrete evidences to show their  hypothesis is validated or disproved.

Reply: Thank you for your professional comments. Comparison A(25/13℃ vs. 13/13℃), Comparison C:(25/13℃ vs. 25/25℃), and Comparison B( 13/13℃vs. 25/25℃). Some differentially expressed genes (DEGs) underwent significant changes only in Comparison A and Comparison C, but not in Comparison B, i.e., the DEGs occurred only in the temperature difference-related Comparison A and C, but not in thecomparison B of constant higher/lower temprature, so we suggest that these DEGs may be involved in the response to DIF,  We added some references ( Lines 469-530, 556-566, Page 20-22) to support the functions of these genes in the discussion, and hypothesized that temperature difference may further affect the function of these DEGs by affecting their expression.We had also corrected inaccuracies in the presentation of the abstract of the manuscript ( Lines 25-26, 30-32, Page 1; Lines 565-569, Page 22), such as‘Therefore, we hypothesize that these genes play an important role in the regulatory network of DIF on functional metabolites of D. officinale PLBs.’had been corrected to‘Therefore, we speculated that these genes may play an important role in the regulatory network of the DIF in the functional metabolites of D. officinale PLBs.’

Comment 4: Figure 1.  What are the sample sizes?  Also, please, state the sample collection detail for example “immediately after 5 day treatment”  

Reply: Thank you for your comments. The samples we used in this study were shown as the following figure, the masses of protocorms are yellow-green, and the sizes are about 10×10 cm,and the prorocrom

Size is 1mm×1mm in diameter.

And the sample collection detail had been added on Lines 575-586 , Page 22 of the manuscript.

Comment 5:Line 130-132: Which cells were used to for RNA-seq?

Reply: Thank you for your comments. In our study, the D.officinale PLBs (Dendrobium officinale)treated under differrence diurnal temperatures (DIFs) (T1: 13/13 °C, T2: 25/13 °C, and T3: 25/25 °C)  were used to for RNA-seq, we had added the description in the revised manuscript (Lines 132-134, Page 4; Lines 623-626, Page 23).

Comment 5: Figure 2B/C: A more informative way to present this data would be to show the enrichment factor.  For example, 20 DEGs found in your comparison would be significant if 10 DEGs are expected, which means 2-factor enrichment.

Reply: We sincerely appreciate the reviewer’s comments.Figure2D is the results of Pathway enrichment of DEGs.We are very sorry for missing the legend of Figure2D. The error had been corrected in the manuscript.In order to make the image clearer, Figure 2 has been divided into three figures: Figure 2, Figure 3, and Figure 4. “Figure 2 MA-plot analysis of DEGs in D. officinale PLBs”; “Figure 3 GO functional classification of DEGs. (A,C,E): the number of DEGs. (B,D,F): the number of up-/down-regulated DEGs.”; “Figure 4 KEGG pathway classification of significantly enriched DEGs. (A,C,E): the pathway classification of DEGs; (B,D,F): Results of Pathway enrichment of differentially expressed genes.” We had also added “Table S1 GO enrichment analysis of DEGs” and “Table S2 The top 20 KEGG pathways enriched by DEGs” to supplement the explanation.

Comment 6: Lines 306-321: The role of HSP family proteins are not clear from this study.  The expression of those genes are changing, which does not mean DIF regulation occurs through them.  These genes might the results of temperature variations, not regulators of this process.

Reply: Thank you for your professional comments. In our study, Comparison A is a comparison of temperature difference between day and night (DIF:25/13℃) and constant low temperature (13/13℃), Comparison C is a comparison of DIF(25/13℃) and constant high temperature (25/25℃), and Comparison B is a comparison of constant low temperature (13/13℃) and constant high temperature (25/25℃).The DEGs showed a significant expression fold change under DIF, especially HSF1 and HSP18.2[Liu, Y., et al., Arabidopsis heat shock factor HsfA1a directly senses heat stress, pH changes, and hydrogen peroxide via the engagement of redox state. Plant Physiology & Biochemistry Ppb, 2013. 64(5): p. 92-98.]. These two genes were most significantly up-regulated only in Comparison A and Comparison C, and the change trend was the same.In the results of qPCR, the change trend of the two genes was the same as the sequencing results of transcriptome. Since HSF1(HSFA1A) has been reported to regulate HSP18.2, which is involved in plant growth, development, flowering, and resistance[Ceylan, Y., Y.C. Altunoglu and E. Horuz, HSF and Hsp Gene Families in sunflower: a comprehensive genome-wide determination survey and expression patterns under abiotic stress conditions. Protoplasma, 2023. 260(6): p. 1473-1491.], so we propose that HSF1 and HSP18.2 may be candidate genes for studying the regulation of plant response to DIF. The effects of DIF on PLBs of D. officinale need to be further verified by our subsequent experiments. Additionally, we preliminarily predicted the potential regulation of these genes through transcriptomic research, and in subsequent experiments, we will conduct functional studies on the candidate genes.

Comment 7:Line 19-23: Move into the main text.

Reply: Thank you for your comments. We had modified it accordingly.

Comment 8:Line 16: What is DIF?

Reply: We had added the full name of DIF when it appeared for the first time(Line 61, Page 2).

Round 2

Reviewer 3 Report

Comments and Suggestions for Authors

The manuscript is improved, however it is still not very well organized.  In the current format, it is very difficult to know if some of pathways suggested by authors are truly DIF-associated or simply temperature-dependent. 

1. It is not clear what the authors wants to test.  I assume that DIF is about specific responses between T1/T2 or T2/T3.

A better way to pick DIF-associated genes would be

1) find DEGs with T1/T2 or  T2/T3 comparisons

2) filter out DEGs from T1/T3 comparison

3) Run pathway enrichment after filtering

For example, many of genes presented in Figure 5 and Figure 7 is largely driven by comparison B, which is not DIF.

2. Also, it would be informative if authors can show the overlap of DEGs among different comparisons.  Something like a Venn diagram, which can visualize the number of genes would be nice.

3. Figure 9: I understand that qPCR results largely backs up the RNA-seq data.  However other explanation seems to support the authors' choice of those genes. 

Comments on the Quality of English Language

A minor checking to have a better flow is highly recommended.

For example, lines 28-30

 In particular, the expression levels of HSP18.2, HSP70, and28
HSFA1A were significantly increased under DIF treatment, which suggest that HSFA1A,HSP7029 and HSP18.2 may respond to the DIF

Author Response

The itemized list of changes addressing reviewer comments

Dear respected editor and reviewers,

Thank you so much for your professional comments on our manuscript entitled ‘Transcriptomic analysis for diurnal temperature differences reveals gene regulation network response to accumulation of bioactive ingredients of protocorm-like bodies in Dendrobium officinale’, which was submitted to the Plants (Basel) (Manuscript ID: plants-2816909). Your valuable opinions helped us a lot to revise our manuscript and improve its quality. According to your comments, we had modified our manuscripts carefully. Now we resubmit the revised manuscript to your journal and hope it would be considered for publication.

The following is a point-to-point response to the reviewer’s comments.

Response to Reviewer

  1. It is not clear what the authors wants to test.  I assume that DIF is about specific responses between T1/T2 or T2/T3.

A better way to pick DIF-associated genes would be

1) find DEGs with T1/T2 or  T2/T3 comparisons

2) filter out DEGs from T1/T3 comparison

3) Run pathway enrichment after filtering

 For example, many of genes presented in Figure 5 and Figure 7 is largely driven by comparison B, which is not DIF.

Reply: Thank you for your professional comments. What we want to study is the effect of temperature differences between day and night (DIF) on D. officinale PLBs.  According to your comments, we had made a modification to identify the genes that respond specifically to DIF and better highlight the purpose of our research. In our revised manuscript, we performed the screening of differentially expressed genes in temperature-specific response accordingly and added "2.3.5 KEGG analysis of DEGs in DIF-specific response" to the manuscript. ( Lines 251-277,Page 10-12) We found that in Comparison A and Comparison C, after filtering out the differentially expressed genes in Comparison B, the pathway enrichment of temperature-specific responses changed a bit compared with the previous enrichment result. DEGs were mainly enriched in pathways related to repairing plant damage and improving resistance, such as Plant-pathogen interaction, DNA replication, Mismatch repair, and Glutathione metabolism. Several pathways involved in secondary metabolites, such as flavonoid biosynthesis, fructose and mannose metabolism and diterpenoid biosynthesis, had also undergone significantly enriched.

We believe that there are also some temperature-responsive genes that do not necessarily not respond in Comparison B. For example, some DEGs may respond with the highest expression under DIF, while differential expression also occurs at high and low temperatures. Therefore, we did not specifically remove the differential genes in Comparison A and Comparison C which were also present in Comparison B. However, in order to correct the problems that present in our previous manuscript, we have modified our presentation. In “2.3.3 Gene ontology (GO) analysis of DEGs” and “2.3.4 Kyoto Encyclopedia of Genes and Genomes pathway (KEGG) analysis”, we ignored Comparison B and focused on DEGs enrichment pathways in Comparison A and Comparison C (Lines 182-194, Page 7; Lines 205-207, Page 8; Lines 212-218, Page 8-9; Lines 224-231, Page 9).

Meanwhile, in order to better represent the response of DEGs in temperature difference in Figure 8 and Figure 10 (Previous figures 5 and 7), we removed the figure display of Comparison B (13/13℃ vs. 25/25℃) ( Figure 8, Lines 302, Page13; Figure 10, Lines 374, Page16). We also deleted the Figure 8 in previous manuscripts and replaced with the heat map “Figure 11 The expression analysis of DEGs in the heat stress pathway.” to represent the DEGs expression among the three comparisons to visually represent the DEGs of DIF-specific response. (Lines 399, Page 17)

In the revised manuscript, we added the analyses of comparison A and C (Lines 288-295, Page 12; Lines 342-344, Page 15). Moreover, we made analysis for some specific response genes in the corresponding schedules (Table S4-7) that are only expressed in Comparison A or C or only in Comparisons A+C but not in Comparison B. (Lines 322-331, Page14; Lines 358-366, Page15; Lines 389-398, Page17).We also discuss these DIF-specific genes.

Details are listed as follow:

  • Lines 251-277,Page 10-12:

2.3.5 KEGG analysis of DEGs in DIF-specific response

The DEGs co-expressed in Comparison B were filtered in Comparison A. The remaining DEGs responded specifically to DIF, especially to DIF during the daytime. These DEGs were subjected to KEGG enrichment analysis, and DNA replication was the most enriched pathway, followed by Mismatch repair, Diterpenoid biosynthesis, Plant-pathogen interaction, Protein export, Fructose and mannose metabolism. (Figure 7A)

The DEGs co-expressed in Comparison B were filtered in Comparison C. The remaining DEGs also responded specifically to DIF, mainly in response to nighttime DIF. These DEGs are mainly enriched in Glutathione metabolis, Fructose and mannose metabolism, Basal transcription factors, Diterpenoid biosynthesis, Stilbenoid, diarylheptanoid and gingerol biosynthesis, MAPK signaling pathway - plant.(Figure 7B)

After integrating the DEGs in Comparison A and Comparison C, and filtering the co-expressed DEGs in Comparison B, the remaining DEGs contained all the DIF-specific response DEGs.The pathways that were significantly enriched for these DEGs were plant-pathogen interaction, followed by DNA replication, mismatch repair, Flavonoid biosynthesis, and fructose and mannose metabolism. These were followed by Basal transcription factors, Protein export. (Figure 7C)

The DIF mainly induced changes in pathways related to plant damage repair and resistance enhancement, such as plant-pathogen interaction, DNA replication, mismatch repair, glutathione metabolism. Some pathways related to functional metabolites such as flavonoid biosynthesis, fructose and mannose metabolism, diterpenoid biosynthesis, etc., had also undergone significant changes.

  • Lines 182-194, Page 7

 In terms of biological processes, the three main functional types with the highest number of DEGs in the DIF comparisons A and C were the same: the metabolic process (676 and 583 genes), cellular process (620 and 561 genes) and single organism process (498 and 461 genes). In the classification of cellular components, the top three numbers of DEGs in Comparison A and Comparison C were in the following aspects: cell (560 and 455 genes), cell part, (558and 451 genes), and membrane (423 and 415 genes). In the classification of molecular function, the top three functional types that enriched the greatest number of DEGs in the DIF comparisons A and C were catalytic activity (713 and 639 genes), binding (614 and 516 genes), and transporter activity (78 and 79 genes) (Figure 5A, C). In conclusion, the GO functional enrichment of DEGs was basically the same in DIF comparisons A and C. However, the number of DEGs in Comparison A was basically greater than that in Comparison C, and more DEGs were down--regulated than up-regulated.(Figure 5B, D).

  • Lines 205-207, Page 8

In two comparisons of DIF, i.e.,Comparisons A and C, there were 3112 and 3322 DEGs that were matched to 131and 130 KEGG metabolic pathways, respectively.

  • Lines 212-218, Page 8-9

In the metabolism branch, there were 2379, and 2510 genes annotated to 11 types of metabolic pathways in DIF comparisons A,and C, respectively. Among them, those involving the highest number of differential genes were all in the global and overview maps.This was followed by the carbohydrate metabolism and, again, the metabolic pathway type, which differed slightly between comparisons (i.e., in Comparison A, it was the amino acid metabolism, while in Comparison C, it was the biosynthesis of other secondary metabolites (Figure 6A,C).

  • Lines 224-231, Page 9

This suggested that these three pathways are responsive to temperature. The three pathways of ABC transporters, Diterpenoid biosynthesis, and ribosome generation in eukaryotes were enriched to the top 20 in both Comparisons A and C. These metabolic pathways are involved in transmembrane material transport, protein synthesis, and alkaloid synthesis in plants. The results indicated that the DIF played an important role in the synthesis of proteins and alkaloids, as well as the transmembrane transport of substances, compared with the whole day constant temperature.

  • Lines 288-301, Page 12

 The top two enriched pathways were all related to DNA synthesis/RNA structure. Histone (DNA. synthesis/ chromatin structure.histone), which can affect plant growth and development, maturation and senescence, and stress response through ubiquitination, methylation, phosphorylation, and acetylation. Gene enrichment occurred in the hormone response annotated to the pathways of ABA metabolism, GA synthesis, late GA synthesis, and jasmonic acid synthesis, which are closely related to plant growth and resistance (Figure S1).

In Comparison C, the DEGs were significantly enriched in the pathways related to secondary metabolism, RNA.regulation of transcription.WRKY domain transcription factor family, hormone response, abiotic stress, lipid metabolism, etc. (Table S3) Therefore, we hypothesised that the DIF caused by night-time temperature changes may cause changes in transcription factors such as WRKY, which in turn regulate the accumulation of secondary metabolites and adaptation to the environment.

  • Lines 342-344, Page 15

The analysis of the DEGs in the secondary metabolism revealed that the difference between the two DIF comparisons, A and C in the pathways of phenlypropanoid, lignin and lignans, flavonoids, terpenoids, carotenoids, and alkaloids-likes (Figure 10).

  • Lines 322-331, Page 14

 In these DEGs, which specifically responded to DIF, HGL1 (dendrobium_glean_10117747) and AMY1 (dendrobium_glean_10026401) changed expression only in Comparison A, suggesting that they may respond to DIF during the day. Starch synthase 3 (SS3, dendrobium_glean_10016137) were found to have significantly up-regulated expression only in Comparison C, indicating that DIF caused by low temperature at night may affect amylose formation. HGL1 (dendrobium_glean_10117749) expression was significantly down-regulated by -1.65 fold and -2.19 fold only in Comparison A and Comparison C, respectively, suggesting that they respond to DIF throughout the day. (Table S5)

  • Lines 358-366, Page15

Expression of GAS2 (dendrobium_glean_10060927), UGT73B4 (dendrobium_glean_10132494), UGT73C1 (dendrobium_glean_10067948) were significantly altered only in Comparison A, suggesting that they may have a specific response to DIF in daytime. UGT73C5 (dendrobium_glean_10022776), GAS2 (dendrobium_glean_10021516, dendrobium_glean_10124444),  UGT73B3 (dendrobium_glean_10052234, dendrobium_glean_10132500), UGT73C6 (dendrobium_glean_10079736) and F3H (dendrobium_glean_10080291) changed expression only in Comparison C, suggesting that they may have a specific response to nocturnal DIF. Expression of some DEGs changed in both Comparisons A and C.

  • Lines 389-398, Page17

 It was found that HSF3 (dendrobium_glean_10092269) and HSFA2 (dendrobium_glean_10039865) were only significantly differentially expressed in Comparisons A and C, thus suggesting that they are involved in the response to the DIF. Among them, the DEGs with the most significant differential expression folds were HSP18.2 (dendrobium_glean_10087095) and HSF1 (dendrobium_glean_ 10092271). Dendrobium_glean_10087095 showed a 9.29- and 4.38-fold increase in expression in Comparisons A and C, respectively. Dendrobium_glean_10092271 showed a 2.99- and 8.55-fold increase in expression in Comparisons A and C, respectively. These results suggested that they may be especially sensitive to the DIF. (Figure 11, Table S7)

  1. Also, it would be informative if authors can show the overlap of DEGs among different comparisons. Something like a Venn diagram, which can visualize the number of genes would be nice.

Reply: Thank you for your professional comments. We added Venn diagram (Figure 4 Lines 170. Page 7) to the analysis the numbers of genes in different comparisons. (Lines 155-165,Page 6) In order to show the Gene expression in each sample, we also added Figure 2 Gene expression in esch sample.(Lines 141,Page 5)

Details are listed as follow:

Lines 155-165,Page 6:

Venn diagram analysis was performed on the differential genes in Comparison A, Comparison B and Comparison C. Among them, 1267 (459 up-regulated/808 down-regulated) DEGs only changed their expression levels in Comparison A, indicating that they were specifically responsive to daytime DIF, while 1304 (620 up-regulated/ 684 down-regulated) DEGs only changed their expression levels in Comparison C, indicating that they were specifically responsive to nighttime DIF. 1348 (496 up-regulated/ 852 down-regulated) DEGs show changes in their expression levels both in Comparisons A and C, indicating that they respond specifically to the DIF throughout the day. In Comparison A and Comparison C, the DEGs repeated in Comparison B were excluded, and there were 3919 DEGs with specific response to DIF (Figure 4).

  1. Figure 9: I understand that qPCR results largely backs up the RNA-seq data. However other explanation seems to support the authors' choice of those genes. 

Reply: Thank you for your professional comments. Based on the transcriptome sequencing results and the main functional product types of D. officinale, we focused on the starch sucrose synthesis pathway, the flavonol pathway, and the heat stress defence pathway. Therefore, when performing qPCR validation, we validated the key genes and significantly differentially expressed genes in these metabolic pathways, including SUS4, SS3, HSP18.2, and HSF1. In addition, we selected some genes with significantly differentially expressed in the enrichment pathways such as pentose phosphate pathway, phenylpropane pathway, lignin synthesis pathway, photosynthesis pathway, and biological clock pathway for validation. These genes include: ferredoxin-like protein, GLDC, ZTL, POP, PFK3, HPPR2, CHS, PSAB, PSBA, COSY, CYP71B24, CRK8, CAD6, ASMT.The results on the one hand validate the accuracy of the transcriptome sequencing results, and on the other hand strongly support the conclusions of our study. This aspect was also described on page 25 and line 686-692 of the manuscript.

Details are listed as follow:

To understand the impact of the temperature on the synthetic pathway at a molecular level, the significantly differentially expressed genes involved in the pathways-including the starch sucrose synthesis pathway, pentose phosphate pathway, phenylpropane pathway, flavonol pathway, heat stress defense, lignin synthesis pathway, photosynthesis, biological clock pathway-were investigated in order to validate the expression difference, and this was achieved using an ABI StepOnePlus Real-Time polymerase chain reaction (PCR) system.

4.Comments on the Quality of English Language

A minor checking to have a better flow is highly recommended.

Reply: We got a professor of this research field to help us polish the language of the manuscript based on the reviewer's comment.

Round 3

Reviewer 3 Report

Comments and Suggestions for Authors

The manuscript has improved significantly, and I support the publication.